# Beyond Unimodal: Generalising Neural Processes for Multimodal Uncertainty Estimation

**Myong Chol Jung**
Monash University
david.jung@monash.edu

**He Zhao**
CSIRO's Data61
he.zhao@ieee.org

**Joanna Dipnall**
Monash University
jo.dipnall@monash.edu

**Lan Du**[*]
Monash University
lan.du@monash.edu

## Abstract

Uncertainty estimation is an important research area to make deep neural networks (DNNs) more trustworthy. While extensive research on uncertainty estimation has been conducted with unimodal data, uncertainty estimation for multimodal data remains a challenge. Neural processes (NPs) have been demonstrated to be an effective uncertainty estimation method for unimodal data by providing the reliability of Gaussian processes with efficient and powerful DNNs. While NPs hold significant potential for multimodal uncertainty estimation, the adaptation of NPs for multimodal data has not been carefully studied. To bridge this gap, we propose Multimodal Neural Processes (MNPs) by generalising NPs for multimodal uncertainty estimation. Based on the framework of NPs, MNPs consist of several novel and principled mechanisms tailored to the characteristics of multimodal data. In extensive empirical evaluation, our method achieves state-of-the-art multimodal uncertainty estimation performance, showing its appealing robustness against noisy samples and reliability in out-of-distribution detection with faster computation time compared to the current state-of-the-art multimodal uncertainty estimation method.

## 1  Introduction

Uncertainty estimation of deep neural networks (DNNs) is an essential research area in the development of reliable and well-calibrated models for safety-critical domains [49, 45]. Despite the remarkable success achieved by DNNs, a common issue with these model is their tendency to make overconfident predictions for both in-distribution (ID) and out-of-distribution (OOD) samples [16, 43, 35]. Extensive research has been conducted to mitigate this problem, but most of these efforts are limited to unimodal data, neglecting the consideration of multimodal data [57, 11, 50, 63, 38, 34].

In many practical scenarios, safety-critical domains typically involve the processing of multimodal data. For example, medical diagnosis classification that uses X-ray, radiology text reports and the patient's medical record history data can be considered as a form of multimodal learning that requires trustworthy predictions [8]. Despite the prevalence of such multimodal data, the problem of uncertainty estimation for multimodal data has not been comprehensively studied. Furthermore, it has been demonstrated that using existing unimodal uncertainty estimation techniques directly for multimodal data is ineffective, emphasising the need for a meticulous investigation of multimodal uncertainty estimation techniques [24, 17].

---

[*]Corresponding author

37th Conference on Neural Information Processing Systems (NeurIPS 2023).

Several studies investigated the problem of multimodal uncertainty estimation with common tasks including 1) assessing calibration performance, 2) evaluating robustness to noisy inputs, and 3) detecting OOD samples. Trusted Multi-view Classification (TMC) [17] is a single-forward deterministic classifier that uses the Demptser's combination rule [7] to combine predictions obtained from different modalities. The current state-of-the-art (SOTA) in this field is Multi-view Gaussian Process (MGP) [24], which is a non-parametric Gaussian process (GP) classifier that utilises the product-of-experts to combine predictive distributions derived from multiple modalities. Although MGP has shown reliability of GPs in this context, it should be noted that the computational cost of a GP increases cubically with the number of samples [19, 65].

Neural processes (NPs) offer an alternative approach that utilises the representation power of DNNs to imitate the non-parametric behaviour of GPs while maintaining a lower computational cost [13, 14]. It has been shown that NPs can provide promising uncertainty estimation for both regression [28, 33, 29] and classification tasks [62, 25] involving unimodal data. Despite the promising potential of NPs for multimodal data, to the best of our knowledge, no research has yet investigated the feasibility of using NPs for multimodal uncertainty estimation.

In this work, we propose a new multimodal uncertainty estimation framework called Multimodal Neural Processes (MNPs) by generalising NPs for multimodal uncertainty estimation. MNPs have three key components: the dynamic context memory (DCM) that efficiently stores and updates informative training samples, the multimodal Bayesian aggregation (MBA) method which enables a principled combination of multimodal latent representations, and the adaptive radial basis function (RBF) attention mechanism that facilitates well-calibrated predictions. Our contributions are:

1. We introduce a novel multimodal uncertainty estimation method by generalising NPs that comprise the DCM, the MBA, and the adaptive RBF attention.

2. We conduct rigorous experiments on seven real-world datasets and achieve the new SOTA performance in classification accuracy, calibration, robustness to noise, and OOD detection.

3. We show that MNPs achieve faster computation time (up to 5 folds) compared to the current SOTA multimodal uncertainty estimation method.

## 2 Background

**Multimodal Classification** The aim of this research is to investigate uncertainty estimation for multimodal classification. Specifically, we consider a multimodal training dataset $D_{train} = \{\{x_i^m\}_{m=1}^M, y_i\}_{i=1}^{N_{train}}$, where $M$ is the number of input modalities and $N_{train}$ is the number of training samples. We assume that each $i^{th}$ sample has $M$ modalities of input $x_i^m \in \mathbb{R}^{d^m}$ with the input dimension of $d^m$ and a single one-hot encoded label $y_i$ with the number of classes $K$. In this study, we consider the input space to be a feature space. The objective is to estimate the labels of test samples $\{y_i\}_{i=1}^{N_{test}}$ given $D_{test} = \{\{x_i^m\}_{m=1}^M\}_{i=1}^{N_{test}}$, where $N_{test}$ is the number of test samples.

**Neural Processes** NPs are stochastic processes using DNNs to capture the ground truth stochastic processes that generate the given data [13]. NPs learn the distribution of functions and provide uncertainty of target samples, preserving the property of GPs. At the same time, NPs exploit function approximation of DNNs in a more efficient manner than GPs. For NPs, both training and test datasets have a context set $C = \{C_X, C_Y\} = \{x_i^C, y_i^C\}_{i=1}^{N_C}$ and a target set $T = (T_X, T_Y) = \{x_i^T, y_i^T\}_{i=1}^{N_T}$ with $N_C$ being the number of context samples and $N_T$ the number of target samples[2], and the learning objective of NPs is to maximise the likelihood of $p(T_Y|C, T_X)$.

Conditional Neural Processes (CNPs), the original work of NPs [13], maximise $p(T_Y|r(C), T_X)$ with $r(C) = \frac{1}{N_C} \sum_{i=1}^{N_C} \text{enc}_\rho(cat\left[x_i^C; y_i^C\right]) \in \mathbb{R}^{d_e}$ where $\text{enc}_\rho$ is an encoder parameterised by $\rho$, $cat[\cdot; \cdot]$ is the concatenation of two vectors along the feature dimension, and $d_e$ is the feature dimension. The mean vector $r(C)$ is the permutation invariant representation that summarises the context set which is passed to a decoder with $T_X$ to estimate $T_Y$.

The original CNPs use the unweighted mean operation to obtain $r(C)$ by treating all the context points equally, which has been shown to be underfitting [28]. To improve over this, Attentive Neural

---

[2]The modality notation is intentionally omitted as the original NPs are designed for unimodal inputs [13, 14].

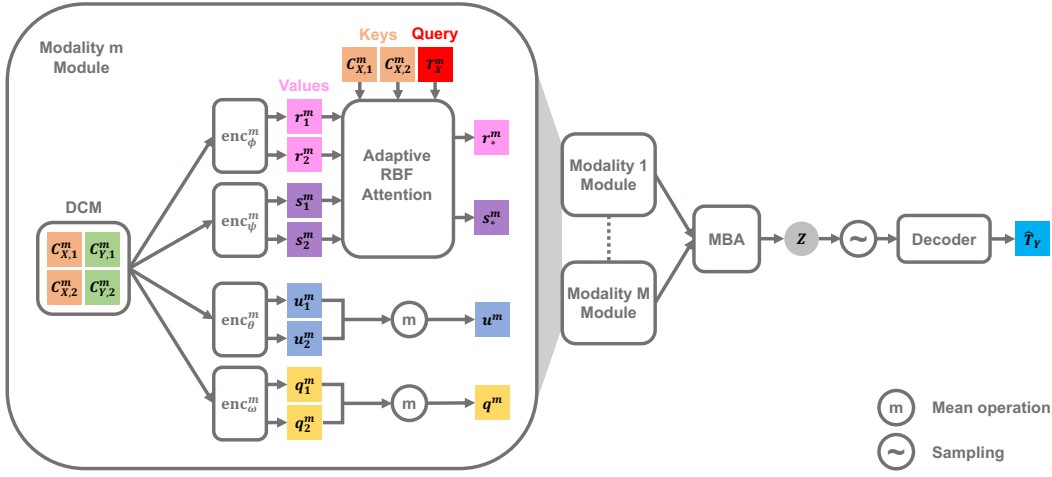

Figure 1: Model diagram of MNPs: DCM refers to Dynamic Context Memory, MBA refers to Multimodal Bayesian Aggregation, and RBF refers to radial basis function.

Processes (ANPs) [28] leveraged the scaled dot-product cross-attention [60] to create target-specific context representations that allocate higher weight on the closer context points:

$$r_*(C, T_X) = \underbrace{\text{Softmax}(T_X C_X^T / \sqrt{d})}_{A(T_X, C_X) \in \mathbb{R}^{N_T \times N_C}} \underbrace{\text{enc}_\rho(cat[C_X; C_Y])}_{\mathbb{R}^{N_C \times d_e}} \tag{1}$$

where $A(T_X, C_X)$ is the attention weight with $T_X$ as the query, $C_X$ is the key, the encoded context set is the value, and $d$ is the input dimension of $T_X$ and $C_X$. The resulted target-specific context representation has been shown to enhance the expressiveness of the task representation [28, 53, 29, 9, 47, 67].

The training procedure of NPs often randomly splits a training dataset to the context and the target sets (i.e., $N_C + N_T = N_{train}$). In the inference stage, a context set is provided for the test/target samples (e.g., a support set in few-shot learning or unmasked patches in image completion). While the context set for the test dataset is given for these tasks, NPs also have been applied to other tasks where the context set is not available (e.g., semi-supervised learning [62] and uncertainty estimation for image classification [25]). In those tasks, existing studies have used a *context memory* during inference, which is composed of training samples that are updated while training [25, 62]. It is assumed that the context memory effectively represents the training dataset with a smaller number of samples. We build upon these previous studies by leveraging the concept of context memory in a more effective manner, which is suitable for our task.

Note that the existing NPs such as ETP [25] can be applied to multimodal classification by concatenating multimodal features into a single unimodal feature. However, in Section 5.1, we show that this approach results in limited robustness to noisy samples.

## 3 Multimodal Neural Processes

The aim of this study is to generalise NPs to enable multimodal uncertainty estimation. In order to achieve this goal, there are significant challenges that need to be addressed first. Firstly, it is essential to have an efficient and effective context memory for a classification task as described in Section 2. Secondly, a systematic approach of aggregating multimodal information is required to provide unified predictions. Finally, the model should provide well-calibrated predictions without producing overconfident estimates as described in Section 1.

Our MNPs, shown in Figure 1, address these challenges with three key components respectively: 1) the dynamic context memory (DCM) (Section 3.1), 2) the multimodal Bayesian aggregation (MBA) (Section 3.2), and 3) the adaptive radial basis function (RBF) attention (Section 3.3). From Section 3.1 to 3.3, we elaborate detailed motivations and proposed solutions of each challenge, and

Section 3.4 outlines the procedures for making predictions. Throughout this section, we refer the multimodal target set $T^M = (\{T_X^m\}_{m=1}^M, T_Y) = \{\{x_i^m\}_{m=1}^M, y_i\}_{i=1}^{N_T}$ to the samples from training and test datasets and the multimodal context memory $C^M = \{C_X^m, C_Y^m\}_{m=1}^M$ to the context set.

## 3.1 Dynamic Context Memory

**Motivation** In NPs, a context set must be provided for a target set, which however is not always possible for non-meta-learning tasks during inference. One simple approach to adapt NPs to these tasks is to randomly select context points from the training dataset, but its performance is suboptimal as randomly sampling a few samples may not adequately represent the entire training distribution. Wang et al. [62] proposed an alternative solution by introducing a first-in-first-out (FIFO) memory that stores the context points with the predefined memory size. Although the FIFO performs slightly better than the random sampling in practice, the performance is still limited because updating the context memory is independent of the model's predictions. Refer to Appendix C.1 for comparisons.

**Proposed Solution** To overcome this limitation of the existing context memory, we propose a simple and effective updating mechanism for the context memory in the training process, which we call Dynamic Context Memory (DCM). We partition context memory $\{C_X^m, C_Y^m\}$ of $m^{th}$ modality into $K$ subsets ($K$ as the number of classes) as $\{C_X^m, C_Y^m\} = \{C_{X,k}^m, C_{Y,k}^m\}_{k=1}^K$ to introduce a class-balance context memory, with each subset devoted to one class. Accordingly, $N^m = N_K^m \times K$ where $N^m$ is the number of context elements per modality (i.e., the size of $\{C_X^m, C_Y^m\}$) and $N_K^m$ is the number of class-specific context samples per modality. This setting resembles the classification setting in [13], where class-specific context representations are obtained for each class.

DCM is initialised by taking $N_K^m$ random training samples from each class of each modality and is updated every mini-batch during training by replacing the least "informative" element in the memory with a possibly more "informative" sample. We regard the element in DCM that receives the smallest attention weight (i.e., $A(T_X, C_X)$ in Equation (1)) during training as the least informative one and the target point that is most difficult to classify (i.e., high classification error) as a more informative sample that should be added to DCM to help the model learn the decision boundary. Formally, this can be written as:

$$C_{X,k}^m[i^*, :] = T_X^m[j^*, :], \quad k \in \{1, \ldots K\}, \quad m \in \{1, \ldots M\} \tag{2}$$

$$i^* = \operatorname*{argmin}_{i \in \{1, \ldots, N_K^m\}} \frac{1}{N_T} \sum_{t=1}^{N_T} \underbrace{A(T_X^m, C_{X,k}^m)}_{\mathbb{R}^{N_T \times N_K^m}}[t, i], \; j^* = \operatorname*{argmax}_{j \in \{1, \ldots, N_T\}} \frac{1}{K} \sum_{k=1}^K \left(T_Y[j, k] - \widehat{T}_Y^m[j, k]\right)^2 \tag{3}$$

where $[i, :]$ indicates the $i^{th}$ row vector of a matrix, $[i, j]$ indicates the $i^{th}$ row and the $j^{th}$ column element of a matrix, and $\widehat{T}_Y^m$ is the predicted probability of the $m^{th}$ modality input. $i^*$ selects the context memory element that receives the least average attention weight in a mini-batch during training, and $j^*$ selects the target sample with the highest classification error. To measure the error between the predictive probability and the ground truth of a target sample, one can use mean squared error (MSE) (Equation (3)) or cross-entropy loss (CE). We empirically found that the former gives better performance in our experiments. Refer to Appendix C.1 for ablation studies comparing the two and the impact of $N^m$ on its performance.

The proposed updating mechanism is very efficient as the predictive probability and the attention weights in Equation (3) are available with no additional computational cost. Also, in comparison to random sampling which requires iterative sampling during inference, the proposed approach is faster in terms of inference wall-clock time since the updated DCM can be used without any additional computation (refer to Appendix C.1 for the comparison).

## 3.2 Multimodal Bayesian Aggregation

**Motivation** With DCM, we obtain the encoded context representations for the $m^{th}$ modality input as follows:

$$r^m = \operatorname{enc}_\phi^m(cat[C_X^m; C_Y^m]) \in \mathbb{R}^{N^m \times d_e}, \quad s^m = \operatorname{enc}_\psi^m(cat[C_X^m; C_Y^m]) \in \mathbb{R}^{N^m \times d_e} \tag{4}$$

where $\phi$ and $\psi$ are the encoders' parameters for the $m^{th}$ modality. Next, given the target set $T_X^m$, we compute the target-specific context representations with the attention mechanism:

$$r_*^m = A(T_X^m, C_X^m)r^m \in \mathbb{R}^{N_T \times d_e}, \quad s_*^m = A(T_X^m, C_X^m)s^m \in \mathbb{R}^{N_T \times d_e} \tag{5}$$

where the attention weight $A(T_X^m, C_X^m) \in \mathbb{R}^{N_T \times N^m}$ can be computed by any scoring function without loss of generality. Note that a single $i^{th}$ target sample consists of multiple representations from $M$ modalities $\{r_{*,i}^m = r_*^m[i,:] \in \mathbb{R}^{d_e}\}_{m=1}^M$. It is important to aggregate these multiple modalities into one latent variable/representation $z_i$ for making a unified prediction of the label.

**Proposed Solution** Instead of using a deterministic aggregation scheme, we propose Multimodal Bayesian Aggregation (MBA), inspired by [61]. Specifically, we view $r_{*,i}^m$ as a sample from a Gaussian distribution with mean $z_i$: $p(r_{*,i}^m|z_i) = \mathcal{N}\left(r_{*,i}^m|z_i, \text{diag}(s_{*,i}^m)\right)$ where $s_{*,i}^m = s_*^m[i,:] \in \mathbb{R}^{d_e}$ is the $i^{th}$ sample in $s_*^m$. Additionally, we impose an informative prior on $z_i$: $p(z_i) = \prod_{m=1}^M \mathcal{N}\left(u^m, \text{diag}(q^m)\right)$ with the mean context representations of $u^m \in \mathbb{R}^{d_e}$ and $q^m \in \mathbb{R}^{d_e}$, which assign uniform weight across the context set as follows:

$$u^m = \frac{1}{N^m} \sum_{i=1}^{N^m} \text{enc}_\theta^m(cat\,[C_X^m[i,:]; C_Y^m[i,:]]), \; q^m = \frac{1}{N^m} \sum_{i=1}^{N^m} \text{enc}_\omega^m(cat\,[C_X^m[i,:]; C_Y^m[i,:]]) \tag{6}$$

where $\theta$ and $\omega$ are the encoders' parameters for the $m^{th}$ modality. Note that the encoders in Equation (4) and (6) are different because they approximate different distribution parameters.

**Lemma 3.1** (Gaussian posterior distribution with factorised prior distribution). *If we have $p(x_i|\mu) = \mathcal{N}(x_i|\mu, \Sigma_i)$ and $p(\mu) = \prod_{i=1}^n \mathcal{N}(\mu_{0,i}, \Sigma_{0,i})$ for $n$ i.i.d. observations of $D$ dimensional vectors, then the mean and covariance of posterior distribution $p(\mu|x) = \mathcal{N}(\mu|\mu_n, \Sigma_n)$ are:*

$$\Sigma_n = \left[\sum_{i=1}^n \left(\Sigma_i^{-1} + \Sigma_{0,i}^{-1}\right)\right]^{-1}, \quad \mu_n = \Sigma_n \left[\sum_{i=1}^n \left(\Sigma_i^{-1}x_i + \Sigma_{0,i}^{-1}\mu_{0,i}\right)\right] \tag{7}$$

As both $p(z_i)$ and $p(r_{*,i}^m|z_i)$ are Gaussian distributions, the posterior of $z_i$ is also a Gaussian distribution: $p(z_i|r_{*,i}) = \mathcal{N}\left(z_i|\mu_{z_i}, \text{diag}(\sigma_{z_i}^2)\right)$, whose mean and variance are obtained by using Lemma 3.1 as follows:

$$\sigma_{z_i}^2 = \left[\sum_{m=1}^M \left((s_{*,i}^m)^\oslash + (q^m)^\oslash\right)\right]^\oslash, \mu_{z_i} = \sigma_{z_i}^2 \otimes \left[\sum_{m=1}^M \left(r_{*,i}^m \otimes (s_{*,i}^m)^\oslash + u^m \otimes (q^m)^\oslash\right)\right] \tag{8}$$

where $\oslash$ and $\otimes$ are element-wise inverse and element-wise product respectively (see Appendix A for proof).

We highlight that if the variance $s_{*,i}^m$ of a modality formed by the target-specific context representation is high in $\sigma_{z_i}^2$, $q^m$ formed by the mean context representations dominates the summation of the two terms. Also, if both $s_{*,i}^m$ and $q^m$ are high for a modality, the modality's contribution to the summation over modalities is low. By doing so, we minimise performance degradation caused by uncertain modalities (see Section 5.1 for its robustness to noisy samples). Refer to Appendix C.2 for ablation studies on different multimodal aggregation methods.

## 3.3 Adaptive RBF Attention

**Motivation** The attention weight $A(T_X^m, C_X^m)$ in Equation (3) and (5) can be obtained by any attention mechanism. The dot-product attention is one of the simple and efficient attention mechanisms, which has been used in various NPs such as [28, 53, 9, 47]. However, we have observed that it is not suitable for our multimodal uncertainty estimation problem, as the dot-product attention assigns excessive attention to context points even when the target distribution is far from the context distribution. The right figure in Figure 2a shows the attention weight in Equation (1) with $A(x_{OOD}^m, C_X^m)$ where $x_{OOD}^m$ is a single OOD target sample (grey sample) far from the context set (red and blue samples). This excessive attention weight results in overconfident predictive probability for OOD samples (see the left figure of Figure 2a), which makes the predictive uncertainty of a classifier hard to distinguish the ID and OOD samples.

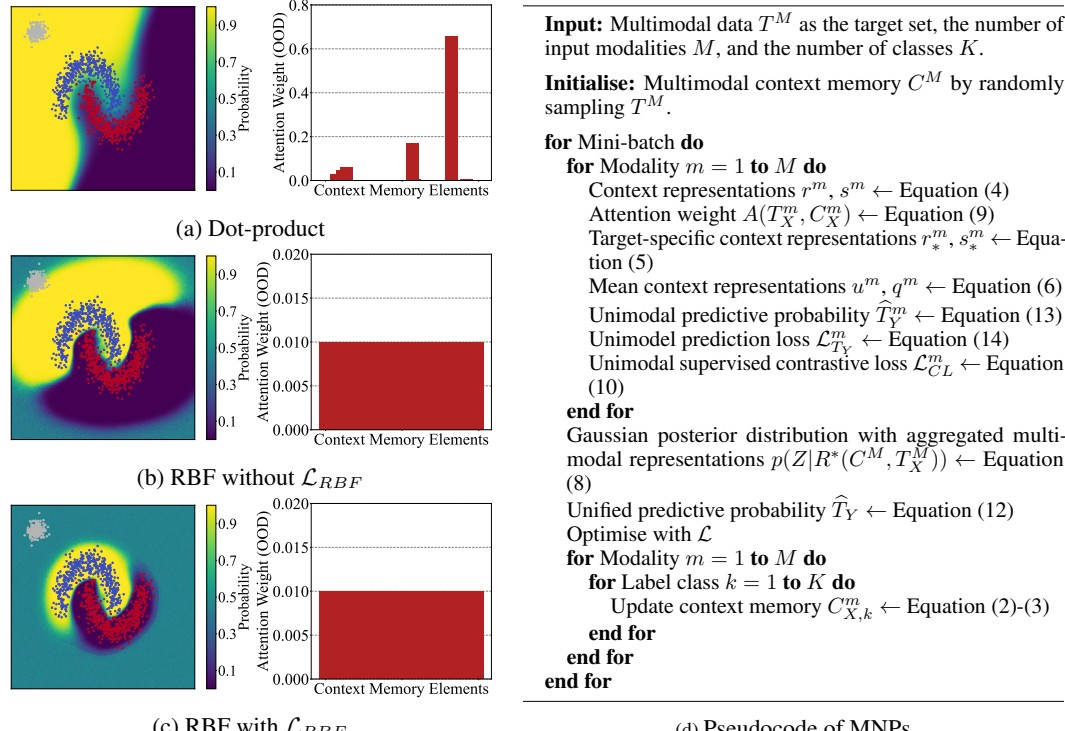

(a) Dot-product

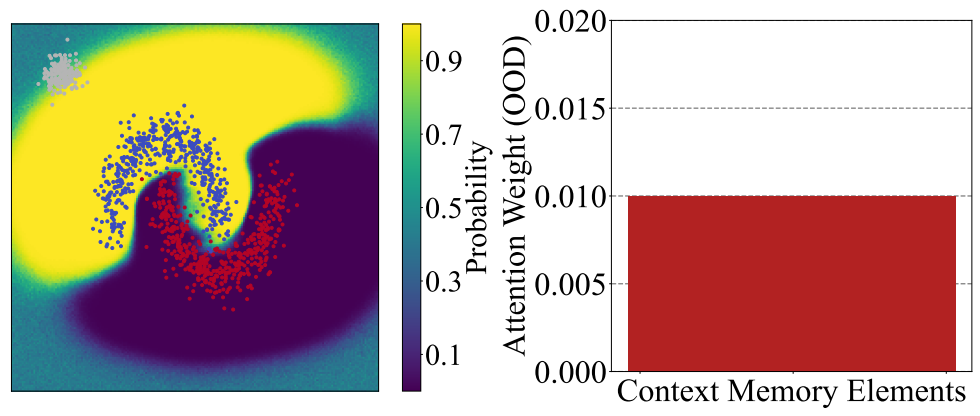

(b) RBF without $\mathcal{L}_{RBF}$

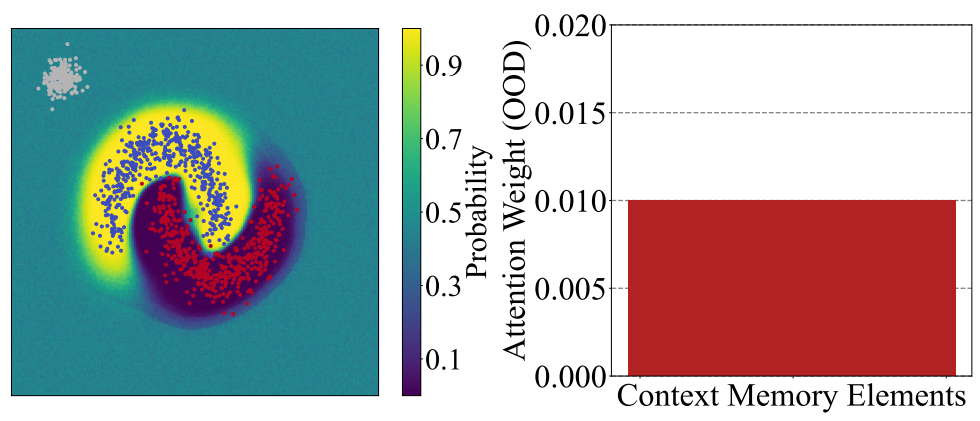

(c) RBF with $\mathcal{L}_{RBF}$

**Input:** Multimodal data $T^M$ as the target set, the number of input modalities $M$, and the number of classes $K$.

**Initialise:** Multimodal context memory $C^M$ by randomly sampling $T^M$.

**for** Mini-batch **do**
  **for** Modality $m = 1$ **to** $M$ **do**
    Context representations $r^m$, $s^m \leftarrow$ Equation (4)
    Attention weight $A(T^m_X, C^m_X) \leftarrow$ Equation (9)
    Target-specific context representations $r^m_*$, $s^m_* \leftarrow$ Equation (5)
    Mean context representations $u^m$, $q^m \leftarrow$ Equation (6)
    Unimodal predictive probability $\widehat{T}^m_Y \leftarrow$ Equation (13)
    Unimodel prediction loss $\mathcal{L}^m_{T_Y} \leftarrow$ Equation (14)
    Unimodal supervised contrastive loss $\mathcal{L}^m_{CL} \leftarrow$ Equation (10)
  **end for**
  Gaussian posterior distribution with aggregated multimodal representations $p(Z|R^*(C^M, T^M_X)) \leftarrow$ Equation (8)
  Unified predictive probability $\widehat{T}_Y \leftarrow$ Equation (12)
  Optimise with $\mathcal{L}$
  **for** Modality $m = 1$ **to** $M$ **do**
    **for** Label class $k = 1$ **to** $K$ **do**
      Update context memory $C^m_{X,k} \leftarrow$ Equation (2)-(3)
    **end for**
  **end for**
**end for**

(d) Pseudocode of MNPs

Figure 2: Predictive probability of upper circle samples (blue) is shown in the left column with (a) the dot-product attention, (b) the RBF attention without $\mathcal{L}_{RBF}$, (c) and the RBF attention with $\mathcal{L}_{RBF}$. Upper circle samples (blue) are class 1, lower circle samples (red) are class 2, and grey samples are OOD samples. The attention weight $A(x^m_{OOD}, C^m_X)$ of an OOD sample across 100 context points is shown in the right column. The summarised steps of training MNPs are shown in (d). Refer to Appendix B for the experimental settings.

**Proposed Solution** To address the overconfident issue of the dot-product attention, we propose an attention mechanism based on RBF. RBF is a stationary kernel function that depends on the relative distance between two points (i.e., $x - x'$) rather than the absolute locations [65], which we define as $\kappa^m(x, x') = \exp\left(-\frac{1}{2}||\frac{x-x'}{(l^m)^2}||^2\right)$ where $||\cdot||^2$ is the squared Euclidean norm, and $l^m$ is the lengthscale parameter that controls the smoothness of the distance in the $m^{th}$ modality input space. The RBF kernel is one of the widely used kernels in GPs [64, 65], and its adaptation with DNNs has shown well-calibrated and promising OOD detection [40, 41, 58, 24]. Formally, we define the attention weight using the RBF kernel as:

$$A(T^m_X, C^m_X) = \text{Sparsemax}(G(T^m_X, C^m_X)) \tag{9}$$

where the elements of $G(T^m_X, C^m_X) \in \mathbb{R}^{N_T \times N^m}$ are $[G]_{ij} = \kappa^m(T^m_X[i,:], C^m_X[j,:])$, and Sparsemax [39] is an alternative activation function to Softmax. It is defined as $\text{Sparsemax}(h) \coloneqq \underset{p \in \Delta^{K-1}}{\text{argmax}} ||p - h||^2$ where $\Delta^{K-1}$ is the $K-1$ dimensional simplex $\Delta^{K-1} \coloneqq \{p \in \mathbb{R}^K | \mathbb{1}^\mathrm{T} p = 1, p \geq \mathbb{0}\}$. Here we use Sparsemax instead of the standard Softmax because Sparsemax allows zero-probability outputs. This property is desirable because the Softmax's output is always positive even when $\kappa^m(x, x') = 0$ (i.e., $||x - x'||^2 \to \infty$) leading to higher classification and calibration errors (see Appendix C.3 for ablation studies).

The lengthscale $l^m$ is an important parameter that determines whether two points are far away from each other (i.e., $\kappa^m(x, x') \to 0$) or close to each other (i.e., $0 < \kappa^m(x, x') \leq 1$). However, in practice, $l^m$ has been either considered as a non-optimisable hyperparameter or an optimisable parameter that requires a complex initialisation [24, 59, 58, 42, 65, 56]. To address this issue, we propose an adaptive learning approach of $l^m$ to form a tight bound to the context distribution by leveraging the supervised contrastive learning [27]. Specifically, we let anchor index $i \in T_{ind} \equiv \{1, \ldots, N_T\}$ with

negative indices $N(i) = T_{ind}\backslash\{i\}$ and positive indices $P(i) = \{p \in N(i) : T_Y[p,:] = T_Y[i,:]\}$. Given an anchor sample of the target set, the negative samples refer to all samples except for the anchor sample, while the positive samples refer to other samples that share the same label as the anchor sample. We define the multimodal supervised contrastive loss as:

$$\mathcal{L}_{CL}^M = \frac{1}{M} \sum_{m=1}^{M} \mathcal{L}_{CL}^m = \frac{1}{M} \sum_{m=1}^{M} \sum_{i=1}^{N_T} -\frac{1}{|P(i)|} \times \sum_{p \in P(i)} \log \frac{\exp\left(\kappa^m(T_X^m[i,:], T_X^m[p,:])/\tau\right)}{\sum_{n \in N(i)} \exp\left(\kappa^m(T_X^m[i,:], T_X^m[n,:])/\tau\right)} \tag{10}$$

where $|P(i)|$ is the cardinality of $P(i)$, and $\tau$ is the temperature scale. This loss encourages higher RBF output of two target samples from the same class and lower RBF output of two target samples from different classes by adjusting the lengthscale. In addition to $\mathcal{L}_{CL}^M$, a $l_2$-loss is added to form the tighter bound by penalising large lengthscale. Overall, the loss term for our adaptive RBF attention is $\mathcal{L}_{RBF} = \mathcal{L}_{CL}^M + \alpha * \frac{1}{M} \sum_{m=1}^{M} ||l^m||$ with the balancing coefficient $\alpha$.

We show the difference between the RBF attention without $\mathcal{L}_{RBF}$ (see Figure 2b) and the adaptive RBF attention with $\mathcal{L}_{RBF}$ (see Figure 2c). It can be seen that the decision boundary modelled by the predictive probability of the adaptive RBF attention is aligned with the data distribution significantly better than the non-adaptive one. For ablation studies with real-world datasets, see Appendix C.4.

## 3.4 Conditional Predictions

We follow the standard procedures of Gaussian process classification to obtain predictions [65, 24, 42, 19] where we first compute the predictive latent distribution $p(f(T_X^M)|C^M, T_X^M)$ as a Gaussian distribution by marginalising $Z = \{z_i\}_{i=1}^{N_T}$:

$$p(f(T_X^M)|C^M, T_X^M) = \int p(f(T_X^M)|Z)p(Z|R^*(C^M, T_X^M))\,dZ \tag{11}$$

where $T_X^M = \{T_X^m\}_{m=1}^M$, $R^*(C^M, T_X^M) = \{r_{*,i}\}_{i=1}^{N_T}$, and $p(f(T_X^M)|Z)$ is parameterised by a decoder. Then, we obtain the predictive probability $\widehat{T}_Y$ by:

$$\widehat{T}_Y = \int \mathrm{Softmax}(p(f(T_X^M)))p(f(T_X^M)|C^M, T_X^M)\,df(T_X^M) \tag{12}$$

Similarly, we can obtain the unimodal predictive latent distribution $p(f(T_X^m)|\{C_X^m, C_Y^m\}, T_X^m)$ and the unimodal predictive probability $\widehat{T}_Y^m$ for the $m^{th}$ modality as:

$$\widehat{T}_Y^m = \int \mathrm{Softmax}(p(f(T_X^m)))p(f(T_X^m)|\{C_X^m, C_Y^m\}, T_X^m)\,df(T_X^m) \tag{13}$$

We minimise the negative log likelihood of the aggregated prediction and the unimodal predictions by:

$$\mathcal{L}_{T_Y} = -\mathbb{E}_{T_Y}\left[\log \widehat{T}_Y\right] - \frac{1}{M} \sum_{m=1}^{M} \mathcal{L}_{T_Y}^m = -\mathbb{E}_{T_Y}\left[\log \widehat{T}_Y\right] - \frac{1}{M} \sum_{m=1}^{M} \mathbb{E}_{T_Y}\left[\log \widehat{T}_Y^m\right] \tag{14}$$

Since Equations (11)-(13) are analytically intractable, we approximate the integrals by the Monte Carlo method [44]. The overall loss for MNPs is $\mathcal{L} = \mathcal{L}_{T_Y} + \beta * \mathcal{L}_{RBF}$ where $\beta$ is the balancing term. Refer to Figure 2d for the summarised steps of training MNPs.

## 4 Related Work

**Neural Processes**  CNP and the latent variant of CNP that incorporates a latent variable capturing global uncertainty were the first NPs introduced in the literature [13, 14]. To address the under-fitting issue caused by the mean context representation in these NPs, Kim et al. [28] proposed to leverage the attention mechanism for target-specific context representations. This approach has been shown to be effective in subsequent works [53, 29, 9, 47, 67, 22]. Many other variants, such as SNP [51, 67], CNAP [48], and MPNPs [5] were proposed for common downstream tasks like 1D regression, 2D image completion [28, 14, 15, 10, 61, 26, 20], and image classification [13, 62, 25]. However, none of these studies have investigated the generalisation of NPs to multimodal data. This study is the first to consider NPs for multimodal classification and its uncertainty estimation.

**Multimodal Learning**   The history of multimodal learning that aims to leverage multiple sources of input can be traced back to the early work of Canonical Correlation Analysis (CCA) [21]. CCA learns the correlation between two variables which was further improved by using feed-forward networks by Deep CCA (DCCA) [2]. With advances in various architectures of DNNs, many studies on multimodal fusion and alignment were proposed [4, 12, 55, 6]. In particular, transformer-based models for vision-language tasks [36, 52, 1] have obtained great attention. Nonetheless, most of these methods were not originally intended for uncertainty estimation, and it has been demonstrated that many of them exhibit inadequate calibration [16, 43].

**Multimodal Uncertainty Estimation**   Multimodal uncertainty estimation is an emerging research area. Its objective is to design robust and calibrated multimodal models. Ma et al. [37] proposed the Mixture of Normal-Inverse Gamma (MoNIG) algorithm that quantifies predictive uncertainty for multimodal regression. However, this work is limited to regression, whereas our work applies to multimodal classification. Han et al. [17] developed TMC based on the Dempster's combination rule to combine multi-view logits. In spite of its simplicity, empirical experiments showed its limited calibration and capability in OOD detection [24]. Jung et al. [24] proposed MGP that combines predictive posterior distributions of multiple GPs by the product of experts. While MGP achieved the current SOTA performance, its non-parametric framework makes it computationally expensive. Our proposed method overcomes this limitation by generalising efficient NPs to imitate GPs.

## 5   Experiments

Apart from measuring the test accuracy, we assessed our method's performance in uncertainty estimation by evaluating its calibration error, robustness to noisy samples, and capability to detect OOD samples. These are crucial aspects that a classifier not equipped to estimate uncertainty may struggle with. We evaluated MNPs on seven real-world datasets to compare the performance of MNPs against four unimodal baselines and three multimodal baselines.

To compare our method against unimodal baselines, we leveraged the early fusion (EF) method [3] that concatenates multimodal input features to single one. The unimodal baselines are (1) **MC Dropout (MCD)** [11] with dropout rate of 0.2, (2) **Deep Ensemble (DE)** [32] with five ensemble models, (3) **SNGP**'s GP layer [35], and (4) **ETP** [25] with memory size of 200 and a linear projector. ETP was chosen as a NP baseline because of its original context memory which requires minimal change of the model to be used for multimodal classification.

The multimodal baselines consist of (1) **Deep Ensemble (DE)** with the late fusion (LF) [3] where a classifier is trained for each modality input, and the final prediction is obtained by averaging predictions from all the modalities, (2) **TMC** [17], and (3) **MGP** [24]. For TMC and MGP, we followed the settings proposed by the original authors. In each experiment, the same feature extractors were used for all baselines. We report mean and standard deviation of results from five random seeds. The bold values are the best results for each dataset, and the underlined values are the second-best ones. See Appendix B for more detailed settings.

### 5.1   Robustness to Noisy Samples

**Experimental Settings**   In this experiment, we evaluated the classification performance of MNPs and the robustness to noisy samples with six multimodal datasets [24, 17]. Following [24] and [17], we normalised the datasets and used a train-test split of 0.8:0.2. To test the robustness to noise, we added zero-mean Gaussian noise with different magnitudes of standard deviation during inference (10 evenly spaced values on a log-scale from $10^{-2}$ to $10^1$) to half of the modalities. For each noise level, all possible combinations of selecting half of the modalities (i.e., $\binom{M}{M/2}$) were evaluated and averaged. We report accuracy and expected calibration error (ECE) for each experiment. Please refer to Appendix B for details of the datasets and metrics.

**Results**   We provide the test results without noisy samples in Table 1, 2, and 5 and the results with noisy samples in Table 3. In terms of accuracy, MNPs outperform all the baselines in 5 out of 6 datasets. At the same time, MNPs provide the most calibrated predictions in 4 out of 6 datasets, preserving the non-parametric GPs' reliability that a parametric model struggles to achieve. This shows that MNPs bring together the best of non-parametric models and parametric models. Also,

Table 1: Test accuracy (↑).

| Method | Handwritten | CUB | PIE | Caltech101 | Scene15 | HMDB |
|--------|-------------|-----|-----|------------|---------|------|
| | | | Dataset | | | |
| MCD | 99.25±0.00 | 92.33±1.09 | 91.32±0.62 | 92.95±0.29 | 71.75±0.25 | 71.68±0.36 |
| DE (EF) | 99.20±0.11 | 93.16±0.70 | 91.76±0.33 | 92.99±0.09 | 72.70±0.39 | 71.67±0.23 |
| SNGP | 98.85±0.22 | 89.50±0.75 | 87.06±1.23 | 91.24±0.46 | 64.68±4.03 | 67.65±1.03 |
| ETP | 98.75±0.25 | 92.33±1.99 | 91.76±0.62 | 92.08±0.33 | 72.58±1.35 | 67.43±0.95 |
| DE (LF) | 99.25±0.00 | 92.33±0.70 | 87.21±0.66 | 92.97±0.13 | 67.05±0.38 | 69.98±0.36 |
| TMC | 98.10±0.14 | 91.17±0.46 | 91.18±1.72 | 91.63±0.28 | 67.68±0.27 | 65.17±0.87 |
| MGP | 98.60±0.14 | 92.33±0.70 | 92.06±0.96 | 93.00±0.33 | 70.00±0.53 | **72.30±0.19** |
| MNPs (Ours) | **99.50±0.00** | **93.50±1.71** | **95.00±0.62** | **93.46±0.32** | **77.90±0.71** | 71.97±0.43 |

Table 2: Test ECE (↓).

| Method | Handwritten | CUB | PIE | Caltech101 | Scene15 | HMDB |
|--------|-------------|-----|-----|------------|---------|------|
| | | | Dataset | | | |
| MCD | 0.009±0.000 | 0.069±0.017 | 0.299±0.005 | 0.017±0.003 | 0.181±0.003 | 0.388±0.004 |
| DE (EF) | 0.007±0.000 | 0.054±0.010 | 0.269±0.004 | 0.036±0.001 | 0.089±0.003 | 0.095±0.003 |
| SNGP | 0.023±0.004 | 0.200±0.010 | 0.852±0.012 | 0.442±0.004 | 0.111±0.063 | 0.227±0.010 |
| ETP | 0.020±0.002 | 0.051±0.009 | 0.287±0.007 | 0.096±0.002 | 0.045±0.008 | 0.100±0.010 |
| DE (LF) | 0.292±0.001 | 0.270±0.009 | 0.567±0.006 | 0.023±0.002 | 0.319±0.005 | 0.270±0.003 |
| TMC | 0.013±0.002 | 0.141±0.002 | 0.072±0.011 | 0.068±0.002 | 0.180±0.004 | 0.594±0.008 |
| MGP | 0.006±0.004 | **0.038±0.007** | 0.079±0.007 | **0.009±0.003** | 0.062±0.006 | 0.036±0.003 |
| MNPs (Ours) | **0.005±0.001** | 0.049±0.008 | **0.040±0.005** | 0.017±0.003 | **0.038±0.009** | **0.028±0.006** |

Table 3: Average test accuracy across 10 noise levels (↑).

| Method | Handwritten | CUB | PIE | Caltech101 | Scene15 | HMDB |
|--------|-------------|-----|-----|------------|---------|------|
| | | | Dataset | | | |
| MCD | 82.15±0.17 | 76.08±0.61 | 64.65±0.77 | 73.45±0.11 | 48.97±0.33 | 42.63±0.08 |
| DE (EF) | 82.16±0.18 | 76.94±0.82 | 65.53±0.20 | 73.99±0.19 | 49.45±0.35 | 41.92±0.06 |
| SNGP | 72.46±0.41 | 61.27±1.24 | 56.52±0.69 | 56.57±0.17 | 38.19±1.86 | 37.49±0.42 |
| ETP | 75.85±0.31 | 73.99±1.12 | 62.64±0.35 | 66.62±0.08 | 46.17±0.58 | 38.28±0.23 |
| DE (LF) | 95.63±0.08 | 76.16±0.28 | 67.69±0.35 | 81.85±0.14 | 50.13±0.27 | 43.01±0.19 |
| TMC | 82.44±0.15 | 74.19±0.69 | 62.18±0.80 | 71.77±0.22 | 42.52±0.29 | 36.61±0.30 |
| MGP | 97.66±0.12 | 85.48±0.25 | 90.97±0.19 | 92.68±0.23 | 65.74±0.56 | **67.02±0.21** |
| MNPs (Ours) | **98.58±0.10** | **88.96±1.98** | **93.80±0.49** | **92.83±0.18** | **74.14±0.35** | 64.11±0.15 |

Table 4: Test accuracy (↑), ECE (↓), and OOD detection AUC (↑).

| Method | Test accuracy ↑ | ECE ↓ | OOD AUC ↑ SVHN | CIFAR100 |
|--------|-----------------|-------|----------------|----------|
| MCD | 74.76±0.27 | 0.013±0.002 | 0.788±0.022 | 0.725±0.014 |
| DE (EF) | 72.95±0.13 | 0.154±0.048 | 0.769±0.008 | 0.721±0.014 |
| SNGP | 61.51±0.30 | 0.020±0.003 | 0.753±0.026 | 0.705±0.024 |
| ETP | 74.42±0.20 | 0.032±0.003 | 0.780±0.011 | 0.695±0.007 |
| DE (LF) | **75.40±0.06** | 0.095±0.001 | 0.722±0.016 | 0.693±0.006 |
| TMC | 72.42±0.05 | 0.108±0.001 | 0.681±0.004 | 0.675±0.006 |
| MGP | 73.30±0.05 | 0.018±0.001 | 0.803±0.007 | 0.748±0.007 |
| MNPs (Ours) | 74.92±0.07 | **0.011±0.001** | **0.872±0.002** | **0.786±0.005** |

Table 5: Speed-up (×folds) measured by the ratio of wall-clock time per epoch of MNPs against MGP.

| Dataset | Training | Testing |
|---------|----------|---------|
| Handwritten | 1.62 | 2.66 |
| CUB | 1.41 | 2.12 |
| PIE | 2.88 | 5.37 |
| Caltech101 | 1.56 | 3.30 |
| Scene15 | 1.89 | 2.63 |
| HMDB | 2.03 | 3.27 |
| CIFAR10-C | 1.17 | 2.16 |

MNPs provide the most robust predictions to noisy samples in 5 out of 6 datasets achieved by the MBA mechanism. Both unimodal and multimodal baselines except MGP show limited robustness with a large performance degradation.

In addition to its superior performance, MNPs, as shown in Table 5, are also faster than the SOTA multimodal uncertainty estimator MGP in terms of wall-clock time per epoch (up to ×5 faster) measured in the identical environment including batch size, GPU, and code libraries (see Appendix B for computational complexity of the two models). This highly efficient framework was made possible

by DCM that stores a small number of informative context points. It also highlights the advantage of using efficient DNNs to imitate GPs, which a non-parametric model like MGP struggles to achieve.

## 5.2 OOD Detection

**Experimental Settings** Following the experimental settings of [24], we trained the models with three different corruption types of CIFAR10-C [18] as a multimodal dataset and evaluated the OOD detection performance using two different test datasets. The first test dataset comprised half CIFAR10-C and half SVHN [46] samples, while the second test dataset comprised half CIFAR10-C and half CIFAR100 [31] samples. We used the Inception v3 [54] pretrained with ImageNet as the backbone of all the baselines. The area under the receiver operating characteristic (AUC) is used as a metric to classify the predictive uncertainty into ID (class 0) and OOD (class 1).

**Results** Table 4 shows test accuracy and ECE with CIFAR10-C and OOD AUC against SVHN and CIFAR100. MNPs outperform all the baselines in terms of ECE and OOD AUC. A large difference in OOD AUC is observed which shows that the proposed adaptive RBF attention identifies OOD samples well. Also, we highlight MNPs outperform the current SOTA MGP in every metric. A marginal difference in test accuracy between DE (LF) and MNPs is observed, but MNPs achieve much lower ECE (approximately 8.6 folds) with higher OOD AUC than DE (LF).

## 6 Conclusion

In this study, we introduced a new multimodal uncertainty estimation method by generalising NPs for multimodal uncertainty estimation, namely Multimodal Neural Processes. Our approach leverages a simple and effective dynamic context memory, a Bayesian method of aggregating multimodal representations, and an adaptive RBF attention mechanism in a holistic and principled manner. We evaluated the proposed method on the seven real-world datasets and compared its performance against seven unimodal and multimodal baselines. The results demonstrate that our method outperforms all the baselines and achieves the SOTA performance in multimodal uncertainty estimation. A limitation of this work is that despite the effectiveness of the updating mechanism of DCM in practive, it is not theoretically guaranteed to obtain the optimal context memory. Nonetheless, our method effectively achieves both accuracy and reliability in an efficient manner. We leave developing a better updating mechanism for our future work. The broader impacts of this work are discussed in Appendix D.

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

# A Lemma and Proof

For the comprehensiveness of proof, we duplicate Lemma 3.1 here.

**Lemma A.1** (Gaussian posterior distribution with factorised prior distribution). *If we have $p(x_i|\mu) = \mathcal{N}(x_i|\mu, \Sigma_i)$ and $p(\mu) = \prod_{i=1}^{n} \mathcal{N}(\mu_{0,i}, \Sigma_{0,i})$ for $n$ i.i.d. observations of $D$ dimensional vectors, then the mean and covariance of posterior distribution $p(\mu|x) = \mathcal{N}(\mu|\mu_n, \Sigma_n)$ are:*

$$\Sigma_n = \left[ \sum_{i=1}^{n} \left( \Sigma_i^{-1} + \Sigma_{0,i}^{-1} \right) \right]^{-1}, \quad \mu_n = \Sigma_n \left[ \sum_{i=1}^{n} \left( \Sigma_i^{-1} x_i + \Sigma_{0,i}^{-1} \mu_{0,i} \right) \right] \tag{15}$$

*Proof.*

$$p(\mu|x) \propto \prod_{i=1}^{n} \frac{1}{\sqrt{(2\pi)^D |\Sigma_i|}} \exp\left( -\frac{1}{2}(x_i - \mu)^T \Sigma_i^{-1}(x_i - \mu) \right) \times$$

$$\frac{1}{\sqrt{(2\pi)^D |\Sigma_{0,i}|}} \exp\left( -\frac{1}{2}(\mu - \mu_{0,i})^T \Sigma_{0,i}^{-1}(\mu - \mu_{0,i}) \right) \tag{16}$$

$$\propto \exp\left[ -\frac{1}{2} \left( \sum_{i=1}^{n} (\mu - x_i)^T \Sigma_i^{-1}(\mu - x_i) + (\mu - \mu_{0,i})^T \Sigma_{0,i}^{-1}(\mu - \mu_{0,i}) \right) \right] \tag{17}$$

$$\propto \exp\left[ -\frac{1}{2} \left( \mu^T \left( \sum_{i=1}^{n} \left( \Sigma_i^{-1} + \Sigma_{0,i}^{-1} \right) \right) \mu - 2\mu^T \left( \sum_{i=1}^{n} \left( \Sigma_i^{-1} x_i + \Sigma_{0,i}^{-1} \mu_{0,i} \right) \right) \right) \right] \tag{18}$$

$$= \frac{1}{\sqrt{(2\pi)^D |\Sigma_n|}} \exp\left( -\frac{1}{2}(\mu - \mu_n)^T \Sigma_n^{-1}(\mu - \mu_n) \right) \tag{19}$$

where we dropped constant terms for clarity. From Equation (18) and (19), we can see that:

$$\Sigma_n^{-1} = \sum_{i=1}^{n} \left( \Sigma_i^{-1} + \Sigma_{0,i}^{-1} \right), \quad \Sigma_n^{-1} \mu_n = \sum_{i=1}^{n} \left( \Sigma_i^{-1} x_i + \Sigma_{0,i}^{-1} \mu_{0,i} \right) \tag{20}$$

$$\Sigma_n = \left[ \sum_{i=1}^{n} \left( \Sigma_i^{-1} + \Sigma_{0,i}^{-1} \right) \right]^{-1}, \quad \mu_n = \Sigma_n \left[ \sum_{i=1}^{n} \left( \Sigma_i^{-1} x_i + \Sigma_{0,i}^{-1} \mu_{0,i} \right) \right] \tag{21}$$

$\square$

If we use Lemma A.1 with diagonal covariance matrices for $p(r_{*,i}^m|z_i) = \mathcal{N}\left( r_{*,i}^m|z_i, \text{diag}(s_{*,i}^m) \right)$ and $p(z_i) = \prod_{m=1}^{M} \mathcal{N}\left( u^m, \text{diag}(q^m) \right)$, we can obtain the posterior distribution of $\mathcal{N}\left( z_i|\mu_{z_i}, \text{diag}(\sigma_{z_i}^2) \right)$ as follows:

$$\sigma_{z_i}^2 = \left[ \sum_{m=1}^{M} \left( (s_{*,i}^m)^\oslash + (q^m)^\oslash \right) \right]^\oslash, \mu_{z_i} = \sigma_{z_i}^2 \otimes \left[ \sum_{m=1}^{M} \left( r_{*,i}^m \otimes (s_{*,i}^m)^\oslash + u^m \otimes (q^m)^\oslash \right) \right] \tag{22}$$

where $\oslash$ is the element-wise inversion, and $\otimes$ is the element-wise product.

# B Experimental Details

In this section, we outline additional details of the experimental settings including the datasets (Appendix B.1), hyperparameters of the models used (Appendix B.2), metrics (Appendix B.3), and a brief analysis of computational complexity of MGP and MNPs (Appendix B.4). For all the experiments, we used the Adam optimiser [30] with batch size of 200 and the Tensorflow framework. All the experiments were conducted on a single NVIDIA GeForce RTX 3090 GPU.

## B.1 Details of Datasets

**Synthetic Dataset** Figure 2 shows the predictive probability and the attention weight of different attention mechanisms. Here, we describe the dataset and the settings used for the demonstrations.

We generated 1,000 synthetic training samples (i.e., $N_{train} = 1,000$) for binary classification by using the Scikit-learn's moon dataset [3] with zero-mean Gaussian noise ($std = 0.15$) added. The test samples were generated as a mesh-grid of 10,000 points (i.e., $100 \times 100$ grid with $N_{test} = 10,000$). The number of points in the context memory $N^m$ was set to 100. In this demonstration, we simplified the problem by setting $M = 1$ which is equivalent to the unimodal setting and illustrated the difference in attention mechanisms.

**Robustness to Noisy Samples Dataset** In Section 5.1, we evaluated the models' robustness to noisy samples with the six multimodal datasets. The details of each dataset are outlined in Table 6. These datasets lie within a feature space where each feature extraction method can be found in [17].

Table 6: Multimodal datasets used for evaluating robustness to noisy samples.

| | Dataset | | | | | |
|---|---|---|---|---|---|---|
| | Handwritten | CUB | PIE | Caltech101 | Scene15 | HMDB |
| # of modalities | 6 | 2 | 3 | 2 | 3 | 2 |
| Types of modalities | Images | Image&Text | Images | Images | Images | Images |
| # of samples | 2,000 | 11,788 | 680 | 8,677 | 4,485 | 6,718 |
| # of classes | 10 | 10 | 68 | 101 | 15 | 51 |

**OOD Detection Dataset** We used CIFAR10-C [18] which consists of corrupted images of CIFAR10 [31]. 15 types of corruptions and five levels of corruption for each type are available for the dataset. Following [24], we used the first three types as multimodal inputs with different levels of corruption (1, 3, and 5).

## B.2 Details of Models

In our main experiments, four unimodal baselines with the early fusion (EF) method [3] (MC Dropout, Deep Ensemble (EF), SNGP, and ETP) and three multimodal baselines with the late fusion (LF) method [3] (Deep Ensemble (LF), TMC, and MGP) were used. In this section, we describe the details of the feature extractors and each baseline.

**Feature Extractors** We used the same feature extractor for all the methods to ensure fair comparisons of the models. For the synthetic dataset, the 2D input points were projected to a high-dimensional space ($d^m = 128$) with a feature extractor that has 6 residual fully connected (FC) layers with the ReLU activation. For the OOD detection experiment, the Inception v3 [54] pretrained with ImageNet was used as the feature extractor. Note that the robustness to noisy samples experiment does not require a separate feature extractor as the dataset is already in a feature space.

**MC Dropout** Monte Carlo (MC) Dropout [11] is a well-known uncertainty estimation method that leverages existing dropout layers of DNNs to approximate Bayesian inference. In our experiments, the dropout rate was set to 0.2 with 100 dropout samples used to make predictions in the inference stage. The predictive uncertainty was quantified based on the original paper [11].

**Deep Ensemble** Deep ensemble [32] is a powerful uncertainty estimation method that trains multiple independent ensemble members. In the case of the unimodal baseline, we employed five ensemble members, whereas for the multimodal baseline, a single classifier was trained independently for each modality input. In both scenarios, the unified predictions were obtained by averaging the predictions from the ensemble members, while the predictive uncertainty was determined by calculating the variance of those predictions.

---

[3] https://scikit-learn.org/stable/modules/generated/sklearn.datasets.make_moons.html

**SNGP** Spectral-normalized Neural Gaussian Process (SNGP) [35] is an effective and scalable uncertainty estimation method that utilises Gaussian process (GP). It consists of a feature extractor with spectral normalisation and a GP output layer. Since we used the identical feature extractor for all the baselines, we only used the GP layer in this work. Following [24], the model's covariance matrix was updated without momentum with $\lambda = \pi/8$ for the mean-field approximation. As the original authors proposed, we quantified the predictive uncertainty based on the Dempster-Shafer theory [7] defined as $u(x) = K/(K + \sum_{k=1}^{K} \exp(\text{logit}_k(x)))$ where $\text{logit}_k(\cdot)$ is the $k^{th}$ class of output logit with the number of classes $K$.

**ETP** Evidential Turing Processes (ETP) [25] is a recent variant of NPs for uncertainty estimation of image classification. Since none of the existing NPs can be directly applied to multimodal data, there are several requirements to utilise them for multimodal classification: 1) a context set in the inference stage (e.g., context memory) and 2) a method of processing multimodal data. ETP was selected due to its inclusion of the original context memory, requiring minimal modifications to be applicable to our task. We used the memory size of 200 and quantified the predictive uncertainty with entropy as proposed by the original paper [25].

**TMC** Trusted Multi-view Classification (TMC) [17] is a simple multimodal uncertainty estimation based on the Subjective logic [23]. We used the original settings of the paper with the annealing epochs of ten for the balancing term. TMC explicitly quantifies its predictive uncertainty based on the Dempster-Shafer theory [7].

**MGP** Multi-view Gaussian Process (MGP) [24] is the current SOTA multimodal uncertainty estimation method that combines predictive posterior distributions of multiple GPs. We used the identical settings of the original paper with the number of inducing points set to 200 and ten warm-up epochs. Its predictive uncertainty was quantified by the predictive variance as proposed by the original paper [24].

**MNPs (Ours)** The encoders and decoder in Multimodal Neural Processes (MNPs) consist of two FC layers with the Leaky ReLU activation [66] after the first FC layer. A normalisation layer is stacked on top of the second FC layer for the encoders. For $\text{enc}_\psi^m(\cdot)$ and $\text{enc}_\omega^m(\cdot)$ that approximate the variance of distributions, we ensure positivity by transforming the outputs as $h_+ = 0.01 + 0.99 * \text{Softplus}(h)$ where $h$ is the raw output from the encoders. $l^m$ of the adaptive RBF attention was initialised as $10 * \mathbb{1} \in \mathbb{R}^{d^m}$, and DCM was initialised by randomly selecting training samples. We used five samples for the Monte Carlo method to approximate the integrals in Equations (11)-(13), which we found enough in practice. Refer to Table 7 for the hyperparameters of MNPs. We provide the impact of $N^m$ on the model performance in Appendix C.1.

Table 7: Hyperparameters of MNPs.

| Parameter | Handwritten | CUB | PIE | Caltech101 | Scene15 | HMDB | CIFAR10-C |
|:---:|:---:|:---:|:---:|:---:|:---:|:---:|:---:|
| | | | | Dataset | | | |
| $N^m$ | 100 | 200 | 300 | 700 | 300 | 400 | 200 |
| $\alpha$ | 1 | 0.03 | 1 | 1 | 0.0001 | 1 | 1 |
| $\beta$ | 1 | 1 | 1 | 1 | 1 | 1 | 1 |
| $\tau$ | 0.25 | 0.01 | 0.1 | 0.01 | 0.5 | 0.01 | 0.01 |

## B.3 Details of Metrics

Apart from test accuracy, we report the expected calibration error (ECE) [16] and the area under the receiver operating characteristic curve (AUC). ECE is defined as:

$$\text{ECE} = \frac{1}{n} \sum_{i=1}^{b} |B_i| |\text{acc}(B_i) - \text{conf}(B_i)|$$

where $n$ is the number of testing samples, $B_i$ is a bin with partitioned predictions with the number of bins $b$, $|B_i|$ is the number of elements in $B_i$, $\text{acc}(B_i)$ is the accuracy of predictions in $B_i$, and

conf($B_i$) is the average predictive confidence in $B_i$. Following [35] and [24], we set $b = 15$. AUC was used for the OOD detection experiment with ground truth labels of class 0 being the ID samples and class 1 being the OOD samples. Each model's predictive uncertainty was used as confidence score to predict whether a test sample is a ID or OD sample.

## B.4 Computational Complexity of MGP and MNPs

In addition to the empirical difference of wall-clock time per epoch in Table 5, we provide computational complexity of the two models in Table 8. We assume that the number of inducing points in MGP equals to the number of context points in MNPs. During training of MNPs, each modality requires a cross-attention ($\mathcal{O}(N^m N_T)$) and a contrastive learning ($\mathcal{O}((N_T)^2)$) that sum to $\mathcal{O}(M(N^m N_T + (N_T)^2))$ with $M$ being the number of modalities, whereas during inference, each modality only requires the cross-attention which results in $\mathcal{O}(M N^m N_T)$.

Table 8: Computational complexity of MGP and MNPs.

|  | Training | Inference |
|---|---|---|
| MGP | $\mathcal{O}(M(N^m)^3)$ | $\mathcal{O}(M(N^m)^3)$ |
| MNPs (Ours) | $\mathcal{O}(M(N^m N_T + (N_T)^2))$ | $\mathcal{O}(M N^m N_T)$ |

# C Ablation Studies

In this section, we analyse MNPs' performance with different settings and show the effectiveness of the proposed framework.

## C.1 Context Memory Updating Mechanisms

We compare the updating mechanism of DCM based on MSE in Equation (2)-(3) with three other baselines: random sampling, first-in-first-out (FIFO) [62], and cross-entropy based (CE). Random sampling bypasses DCM and randomly selects training samples during inference. For FIFO, we follow the original procedure proposed by [62] that updates the context memory during training and only uses it during inference. CE-based mechanism replaces $j^*$ in Equation (3) with $j^* = \underset{j \in \{1,...,N_T\}}{\mathrm{argmax}} \frac{1}{K} \sum_{k=1}^{K} -T_Y[j,k] \log (\widehat{T}_Y^m[j,k])$.

We provide experimental results for all the experiments outlined in Section 5. We highlight that random sampling and FIFO achieve high accuracy both without noise and with noise as shown in Table 9 and 11. However, MSE and CE outperform the others in terms of ECE in Table 10 and OOD AUC in Table 12. As MSE and CE select the new context points based on classification error, the selected context points tend to be close to decision boundary, which is the most difficult region to classify. We believe this may contribute to the lower calibration error, suppressing overconfident predictions. The MSE and CE mechanisms show comparable overall results, but we selected MSE for its lower ECE. In terms of time efficiency, Table 13 shows that random sampling is slower than the other three methods.

For DCM updated by MSE, we also provide difference in performance for a range of number of context points $N^m$ in Figure 3-9. For every figure, the bold line indicates the mean value, and the shaded area indicates 95% confidence interval. Unsurprisingly, the training time and the testing time increase with respect to $N^m$. The general trend in test accuracy across the datasets shows the benefit of increasing the number of context points. However, the performance gain in ECE and OOD AUC is ambivalent as different patterns are observed for different datasets. We leave an in-depth analysis of this behaviour for our future study.

Table 9: Test accuracy with different context memory updating mechanisms (↑).

| Updating Mechanism | Dataset | | | | | |
|---|---|---|---|---|---|---|
| | Handwritten | CUB | PIE | Caltech101 | Scene15 | HMDB |
| Random | 99.40±0.14 | 88.50±5.12 | 94.85±0.90 | 90.38±1.38 | 76.03±2.96 | 68.42±0.53 |
| FIFO | 99.30±0.11 | 90.33±3.26 | **95.29±1.85** | 91.09±0.97 | 76.08±1.92 | 69.65±0.66 |
| CE | 99.40±0.14 | **93.67±2.25** | 95.00±1.43 | **93.59±0.27** | 77.40±0.73 | 70.77±1.11 |
| MSE | **99.50±0.00** | 93.50±1.71 | 95.00±0.62 | 93.46±0.32 | **77.90±0.71** | **71.97±0.43** |

Table 10: Test ECE with different context memory updating mechanisms (↓).

| Updating Mechanism | Dataset | | | | | |
|---|---|---|---|---|---|---|
| | Handwritten | CUB | PIE | Caltech101 | Scene15 | HMDB |
| Random | 0.007±0.001 | 0.069±0.029 | 0.050±0.009 | 0.043±0.005 | 0.059±0.061 | 0.052±0.006 |
| FIFO | 0.007±0.001 | 0.067±0.021 | 0.057±0.016 | 0.027±0.004 | 0.056±0.048 | 0.032±0.007 |
| CE | 0.006±0.001 | 0.050±0.016 | 0.041±0.009 | **0.017±0.003** | **0.038±0.010** | 0.034±0.008 |
| MSE | **0.005±0.001** | **0.049±0.008** | **0.040±0.005** | **0.017±0.003** | **0.038±0.010** | **0.028±0.006** |

Table 11: Average test accuracy across 10 noise levels with different context memory updating mechanisms (↑).

| Updating Mechanism | Dataset | | | | | |
|---|---|---|---|---|---|---|
| | Handwritten | CUB | PIE | Caltech101 | Scene15 | HMDB |
| Random | 98.39±0.21 | 83.11±4.08 | 92.55±0.55 | 89.36±1.18 | 72.85±2.30 | 62.10±0.44 |
| FIFO | 98.51±0.11 | 85.86±2.87 | **93.81±0.67** | 89.59±1.02 | 72.59±1.82 | 63.00±0.89 |
| CE | 98.49±0.13 | 88.80±1.57 | 93.75±0.72 | **92.87±0.21** | 73.98±0.41 | 63.97±0.71 |
| MSE | **98.58±0.10** | **88.96±1.98** | 93.80±0.49 | 92.83±0.18 | **74.14±0.35** | **64.11±0.15** |

Table 12: Test accuracy (↑), ECE (↓), and OOD detection AUC (↑) with different context memory updating mechanisms.

| Updating Mechanism | Test accuracy ↑ | ECE ↓ | OOD AUC ↑ | |
|---|---|---|---|---|
| | | | SVHN | CIFAR100 |
| Random | 74.61±0.22 | 0.073±0.005 | 0.860±0.003 | 0.777±0.002 |
| FIFO | 74.82±0.11 | 0.073±0.006 | 0.862±0.007 | 0.778±0.005 |
| CE | 74.70±0.19 | 0.013±0.002 | 0.871±0.004 | **0.789±0.004** |
| MSE | **74.92±0.07** | **0.011±0.001** | **0.872±0.002** | 0.786±0.005 |

Table 13: Wall-clock inference time (ms/epoch) with different context memory updating mechanisms.

| Updating Mechanism | Dataset | | | | | | |
|---|---|---|---|---|---|---|---|
| | Handwritten | CUB | PIE | Caltech101 | Scene15 | HMDB | CIFAR10-C |
| Random | 31.80±3.68 | 8.15±2.80 | 12.14±3.09 | 255.37±13.25 | 33.73±3.56 | 79.19±5.95 | 710.48±8.58 |
| FIFO | 24.91±0.68 | 5.87±3.06 | 7.20±2.77 | 101.02±2.90 | **25.04±3.35** | **41.50±2.74** | 496.23±10.85 |
| CE | 25.00±0.28 | 5.61±1.56 | 6.85±1.04 | 101.10±2.59 | 25.47±3.77 | 43.45±3.85 | 500.79±7.04 |
| MSE | **22.53±1.88** | **5.57±1.58** | **6.70±0.92** | **101.01±2.38** | 26.60±10.37 | 41.87±2.10 | **493.18±9.91** |

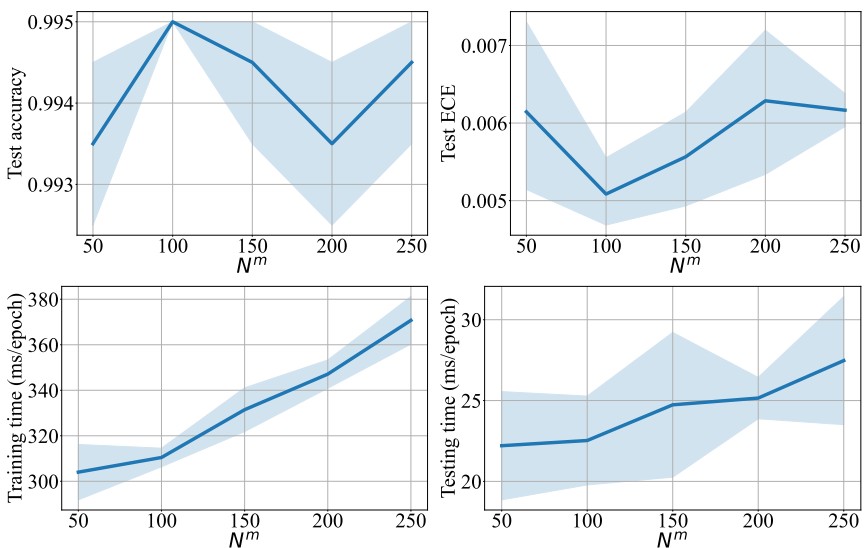

Figure 3: Test accuracy, ECE, average training time, and average testing time with different $N^m$ for the Handwritten dataset.

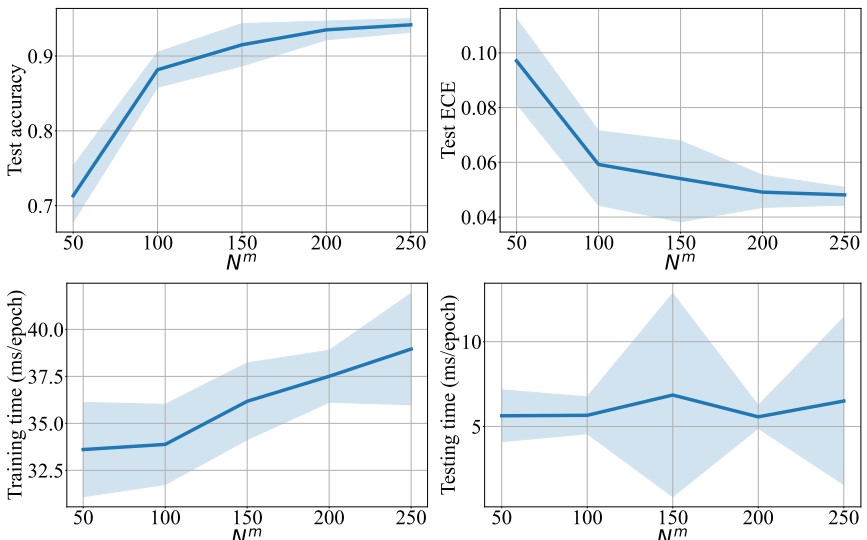

Figure 4: Test accuracy, ECE, average training time, and average testing time with different $N^m$ for the CUB dataset.

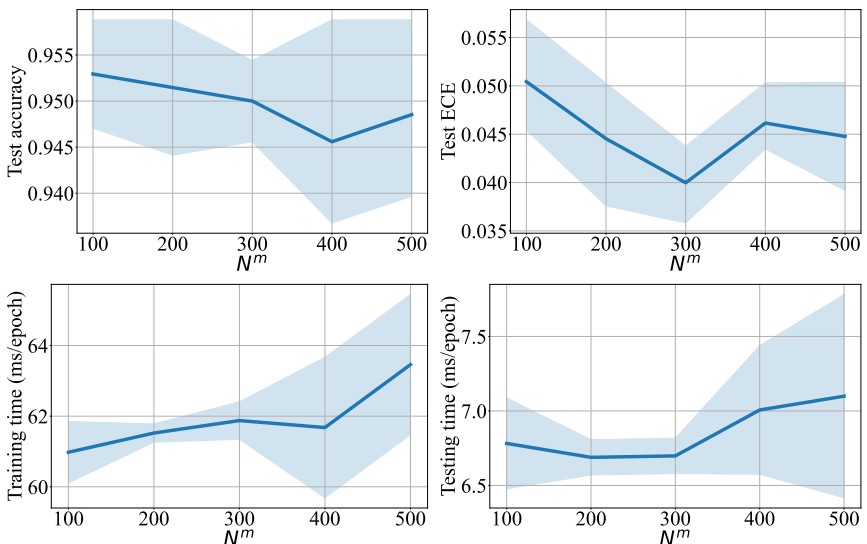

Figure 5: Test accuracy, ECE, average training time, and average testing time with different $N^m$ for the PIE dataset.

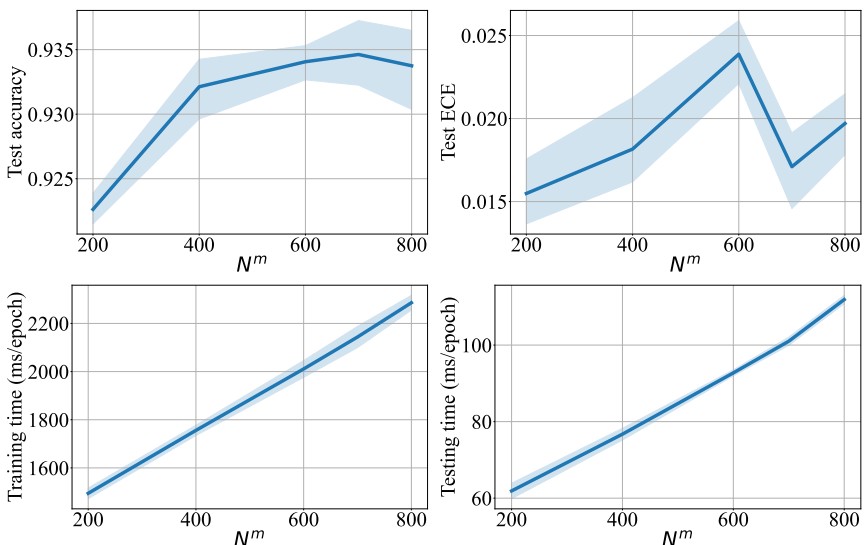

Figure 6: Test accuracy, ECE, average training time, and average testing time with different $N^m$ for the Caltech101 dataset.

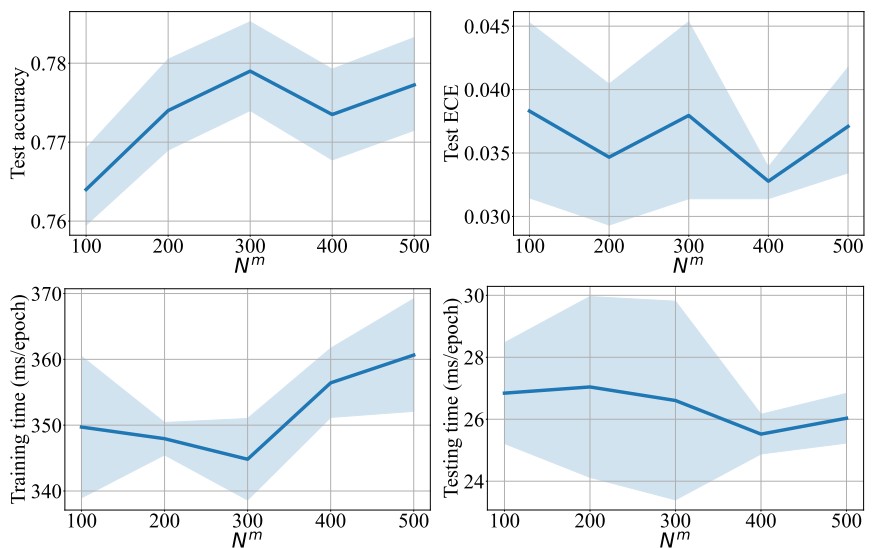

Figure 7: Test accuracy, ECE, average training time, and average testing time with different $N^m$ for the Scene15 dataset.

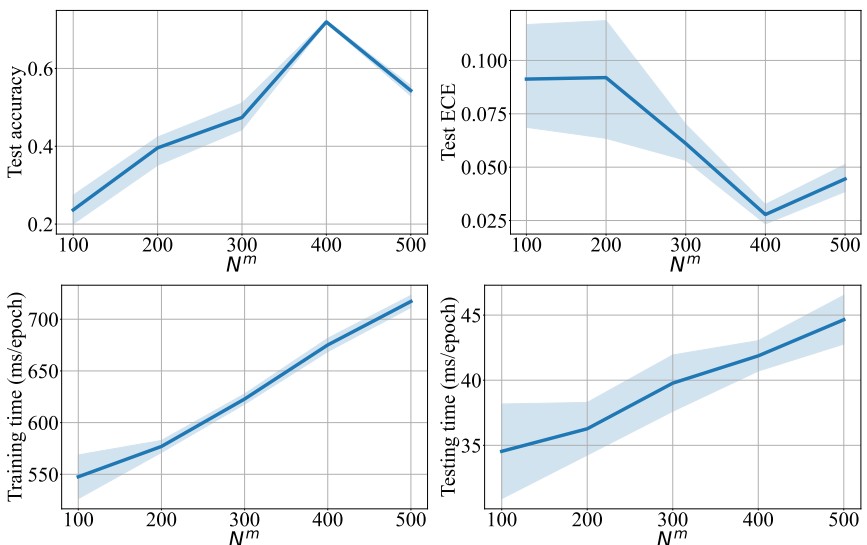

Figure 8: Test accuracy, ECE, average training time, and average testing time with different $N^m$ for the HMDB dataset.

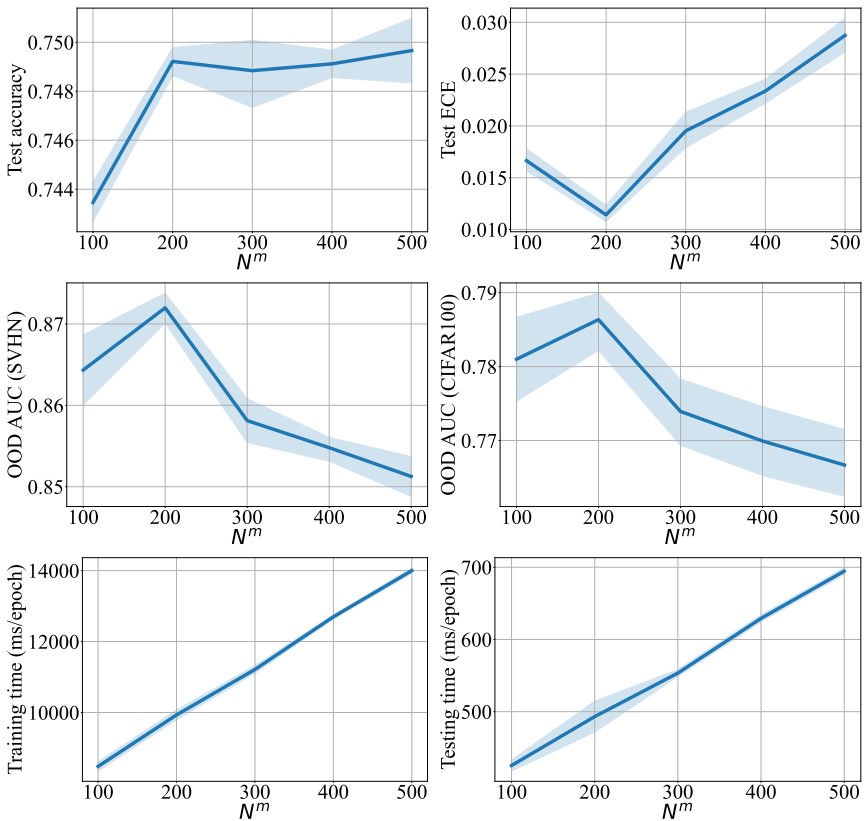

Figure 9: Test accuracy, ECE, OOD AUC (SVHN), OOD AUC (CIFAR100), average training time, and average testing time with different $N^m$ for the CIFAR10-C dataset.

## C.2 Multimodal Aggregation Methods

We demonstrate the performance of MBA compared with two other methods namely "Concat" and "Mean". "Concat" bypasses MBA and directly provides $r_*^m$ of multiple modalities to the decoder (see Figure 1) by simple concatenation followed by passing to a MLP which lets $p(f(T_X^M)|C^M, T_X^M)$ in Equation (12) be parameterised by a decoder where $\{C^M, T_X^M\} = MLP(Concat(\{r_*^m\}_{m=1}^M))$. $Concat(\cdot)$ represents concatenating multiple vectors along their feature dimension. Similarly, "Mean" also bypasses MBA and simply averages the multiple modalities into single representation. Formally, $p(f(T_X^M)|C^M, T_X^M)$ parameterised by a decoder where $\{C^M, T_X^M\} = \frac{1}{M}\sum_{m=1}^M r_*^m$.

The results are shown in Table 14-17. In every case, MBA outperforms both baselines. While similar performance can be observed for Handwritten, Scene15, and Caltech101, large differences are observed in CUB, PIE, and HMDB across different metrics. The test accuracy of CIFAR10 is almost consistent across all methods, but large gaps in ECE and OOD performance are observed. This highlights the importance of MBA, especially in robustness and calibration performance.

Table 14: Test accuracy with different multimodal aggregation methods ($\uparrow$).

| Aggregation Methods | Dataset | | | | | |
|---|---|---|---|---|---|---|
| | Handwritten | CUB | PIE | Caltech101 | Scene15 | HMDB |
| Concat | 99.35±0.22 | 89.00±1.24 | 89.71±2.49 | 92.63±0.18 | 77.18±0.64 | 56.06±2.13 |
| Mean | 99.45±0.11 | 92.50±2.43 | 90.88±2.24 | 93.14±0.25 | 77.60±0.56 | 57.80±1.97 |
| MBA | **99.50±0.00** | **93.50±1.71** | **95.00±0.62** | **93.46±0.32** | **77.90±0.71** | **71.97±0.43** |

Table 15: Test ECE with different multimodal aggregation methods ($\downarrow$).

| Aggregation Methods | Dataset | | | | | |
|---|---|---|---|---|---|---|
| | Handwritten | CUB | PIE | Caltech101 | Scene15 | HMDB |
| Concat | 0.007±0.001 | 0.109±0.008 | 0.092±0.020 | 0.038±0.005 | 0.061±0.005 | 0.060±0.017 |
| Mean | 0.006±0.001 | 0.057±0.012 | 0.059±0.008 | 0.030±0.004 | 0.038±0.005 | 0.117±0.014 |
| MBA | **0.005±0.001** | **0.049±0.008** | **0.040±0.005** | **0.017±0.003** | **0.038±0.009** | **0.028±0.006** |

Table 16: Average test accuracy across 10 noise levels with different multimodal aggregation methods ($\uparrow$).

| Aggregation Methods | Dataset | | | | | |
|---|---|---|---|---|---|---|
| | Handwritten | CUB | PIE | Caltech101 | Scene15 | HMDB |
| Concat | 97.71±0.46 | 85.51±1.42 | 85.94±2.48 | 89.84±0.17 | 72.23±0.52 | 45.22±2.86 |
| Mean | 98.42±0.09 | 88.27±1.83 | 88.74±2.33 | 92.07±0.16 | 74.06±0.28 | 49.58±2.24 |
| MBA | **98.58±0.10** | **88.96±1.98** | **93.80±0.49** | **92.83±0.18** | **74.14±0.35** | **64.11±0.15** |

Table 17: Test accuracy ($\uparrow$), ECE ($\downarrow$), and OOD detection AUC ($\uparrow$) with different multimodal aggregation methods.

| Aggregation Methods | | | OOD AUC $\uparrow$ | |
|---|---|---|---|---|
| | Test accuracy $\uparrow$ | ECE $\downarrow$ | SVHN | CIFAR100 |
| Concat | 74.24±0.27 | 0.125±0.005 | 0.781±0.016 | 0.728±0.004 |
| Mean | 74.72±0.24 | 0.109±0.003 | 0.803±0.007 | 0.742±0.003 |
| MBA | **74.92±0.07** | **0.011±0.001** | **0.872±0.002** | **0.786±0.005** |

## C.3 Attention Types

We decompose the attention weight $A(T_X^m, C_X^m)$ in Equation (9) as follows:

$$A(T_X^m, C_X^m) = \text{Norm}(\text{Sim}(T_X^m, C_X^m)) \tag{23}$$

where $\text{Norm}(\cdot)$ is the normalisation function such as Softmax and Sparsemax, and $\text{Sim}(\cdot, \cdot)$ as the similarity function such as the dot-product and the RBF kernel. We provide experimental results of four different combinations of normalisation functions and similarity functions in Table 18-21.

Among the four combinations, the RBF function with Sparsemax outperforms the others in most cases. More importantly, Table 20 shows a large difference in robustness to noisy samples between the RBF function with Sparsemax and the dot-product with Sparsemax, even when a marginal difference in accuracy is shown in Table 18. For instance, for the PIE dataset, the difference in accuracy without noisy samples is 0.3, but the difference increases to 6.0 in the presence of noisy samples. The same pattern is observed with OOD AUC in Table 21. This illustrates the strength of RBF attention that is more sensitive to distribution-shift as shown in Figure 2. Lastly, for both similarity functions, Sparsemax results in superior overall performance.

Table 18: Test accuracy with different attention mechanisms ($\uparrow$).

| Similarity Function | Normalisation Function | Dataset | | | | | |
|---|---|---|---|---|---|---|---|
| | | Handwritten | CUB | PIE | Caltech101 | Scene15 | HMDB |
| RBF | Softmax | 98.80±0.45 | 87.00±6.42 | 75.15±3.00 | 82.95±0.47 | 69.83±1.41 | 56.28±1.18 |
| | Sparsemax | **99.50±0.00** | **93.50±1.71** | **95.00±0.62** | **93.46±0.32** | 77.90±0.71 | **71.97±0.43** |
| Dot | Softmax | 99.00±0.18 | 79.67±3.94 | 86.32±2.88 | 88.90±0.36 | 74.95±0.33 | 64.68±0.78 |
| | Sparsemax | 98.95±0.11 | 82.17±2.67 | 94.26±1.90 | 92.46±0.26 | **78.30±1.06** | 63.23±1.89 |

Table 19: Test ECE with different attention mechanisms ($\downarrow$).

| Similarity Function | Normalisation Function | Dataset | | | | | |
|---|---|---|---|---|---|---|---|
| | | Handwritten | CUB | PIE | Caltech101 | Scene15 | HMDB |
| RBF | Softmax | 0.019±0.005 | 0.084±0.020 | 0.100±0.017 | 0.025±0.004 | 0.152±0.007 | 0.202±0.019 |
| | Sparsemax | **0.005±0.001** | **0.049±0.008** | **0.040±0.005** | **0.017±0.003** | 0.038±0.009 | **0.028±0.006** |
| Dot | Softmax | 0.008±0.003 | 0.166±0.015 | 0.373±0.037 | 0.033±0.007 | 0.061±0.010 | 0.175±0.006 |
| | Sparsemax | 0.010±0.001 | 0.131±0.028 | 0.053±0.010 | 0.025±0.002 | **0.032±0.008** | 0.084±0.015 |

Table 20: Average test accuracy with different attention mechanisms ($\uparrow$).

| Similarity Function | Normalisation Function | Dataset | | | | | |
|---|---|---|---|---|---|---|---|
| | | Handwritten | CUB | PIE | Caltech101 | Scene15 | HMDB |
| RBF | Softmax | 94.56±0.66 | 82.58±5.98 | 65.88±2.98 | 81.23±0.29 | 67.77±1.05 | 38.63±0.63 |
| | Sparsemax | **98.58±0.10** | **88.96±1.98** | **93.80±0.49** | **92.83±0.18** | **74.14±0.35** | **64.11±0.15** |
| Dot | Softmax | 77.99±0.32 | 73.89±1.77 | 70.80±1.71 | 63.80±0.12 | 58.74±0.24 | 34.28±0.45 |
| | Sparsemax | 96.00±0.24 | 70.30±2.61 | 87.44±1.44 | 81.95±1.92 | 67.84±1.00 | 40.26±0.56 |

Table 21: Test accuracy ($\uparrow$), ECE ($\downarrow$), and OOD detection AUC ($\uparrow$) with different attention mechanisms.

| Similarity Function | Normalisation Function | | | OOD AUC $\uparrow$ | |
|---|---|---|---|---|---|
| | | Test accuracy $\uparrow$ | ECE $\downarrow$ | SVHN | CIFAR100 |
| RBF | Softmax | 67.65±0.16 | 0.080±0.001 | 0.864±0.006 | 0.771±0.006 |
| | Sparsemax | 74.92±0.07 | **0.011±0.001** | **0.872±0.002** | **0.786±0.005** |
| Dot | Softmax | 68.81±0.62 | 0.130±0.019 | 0.849±0.009 | 0.775±0.005 |
| | Sparsemax | **75.07±0.09** | 0.055±0.001 | 0.837±0.004 | 0.765±0.004 |

## C.4 Adaptive Learning of RBF Attention

We have shown that the effectiveness of learning the RBF attention's parameters with the synthetic dataset in Figure 2. We further provide the ablation studies with the real-world datasets in Table 22-25.

Table 22: Test accuracy with and without $\mathcal{L}_{RBF}$ ($\uparrow$).

| Method | Dataset | | | | | |
|---|---|---|---|---|---|---|
| | Handwritten | CUB | PIE | Caltech101 | Scene15 | HMDB |
| Without $\mathcal{L}_{RBF}$ | 96.85±0.29 | 91.17±2.40 | 93.38±1.27 | 92.64±0.38 | 74.45±0.45 | 48.95±1.70 |
| With $\mathcal{L}_{RBF}$ | **99.50±0.00** | **93.50±1.71** | **95.00±0.62** | **93.46±0.32** | **77.90±0.71** | **71.97±0.43** |

Table 23: Test ECE with and without $\mathcal{L}_{RBF}$ ($\downarrow$).

| Method | Dataset | | | | | |
|---|---|---|---|---|---|---|
| | Handwritten | CUB | PIE | Caltech101 | Scene15 | HMDB |
| Without $\mathcal{L}_{RBF}$ | 0.007±0.001 | 0.078±0.011 | 0.043±0.007 | 0.036±0.004 | 0.054±0.011 | 0.043±0.008 |
| With $\mathcal{L}_{RBF}$ | **0.005±0.001** | **0.049±0.008** | **0.040±0.005** | **0.017±0.003** | **0.038±0.010** | **0.028±0.006** |

Table 24: Average test accuracy across 10 noise levels with and without $\mathcal{L}_{RBF}$ ($\uparrow$).

| Method | Dataset | | | | | |
|---|---|---|---|---|---|---|
| | Handwritten | CUB | PIE | Caltech101 | Scene15 | HMDB |
| Without $\mathcal{L}_{RBF}$ | 89.44±0.54 | 86.69±1.65 | 91.50±0.94 | 92.32±0.27 | 71.18±0.38 | 37.33±0.92 |
| With $\mathcal{L}_{RBF}$ | **98.58±0.10** | **88.96±1.98** | **93.80±0.49** | **92.83±0.18** | **74.14±0.35** | **64.11±0.15** |

Table 25: Test accuracy ($\uparrow$), ECE ($\downarrow$), and OOD detection AUC ($\uparrow$) with and without $\mathcal{L}_{RBF}$.

| Method | Test accuracy $\uparrow$ | ECE $\downarrow$ | OOD AUC $\uparrow$ | |
| | | | SVHN | CIFAR100 |
| --- | --- | --- | --- | --- |
| Without $\mathcal{L}_{RBF}$ | **74.96$\pm$0.16** | 0.019$\pm$0.002 | 0.822$\pm$0.004 | 0.746$\pm$0.004 |
| With $\mathcal{L}_{RBF}$ | 74.92$\pm$0.07 | **0.011$\pm$0.001** | **0.872$\pm$0.002** | **0.786$\pm$0.005** |

## D    Broader Impacts

As a long-term goal of this work is to make multimodal classification of DNNs more trustworthy by using NPs, it has many potential positive impacts to our society. Firstly, with transparent and calibrated predictions, more DNNs can be deployed to safety-critical domains such as medical diagnosis. Secondly, this work raises awareness to the machine learning society to evaluate and review reliability of a DNN model. Lastly, our study shows the potential capability of NPs in more diverse applications. Nevertheless, a potential negative impact may exist if the causes of uncertain predictions are not fully understood. To take a step further to reliable DNNs, the source of uncertainty should be transparent to non-experts.

