# OpenReview forum: "Beyond Unimodal: Generalising Neural Processes for Multimodal Uncertainty Estimation"
_NeurIPS.cc/2023/Conference — NeurIPS 2023 poster_

### Official Review · Reviewer_hoCV · 2023-06-27

**Soundness:** 3 good
**Presentation:** 3 good
**Contribution:** 3 good
**Rating:** 6
**Confidence:** 4

**Summary:**

The paper tackles the problem of uncertainty estimation specifically for multi-modal data, i.e., inputs consisting of different sources. Specifically, it improves the popular Neural Process (NP) in three aspects: dynamic context update, multi-modal Bayesian aggregation, and a novel attention mechanism based on the RBF kernel. The paper demonstrates superior test accuracy, uncertainty estimation (calibration and OOD detection), and robustness on multiple benchmarks compared to prior works.

**Strengths:**

* The paper studies the less well-studied area of uncertainty estimation for multi-modal data. This has practical implications as many real-world applications can be multi-modal.
* The paper presents multiple metrics including accuracy, calibration, OOD detection, and robustness, which provide a holistic evaluation of the proposed method.
* The idea of dynamic context update is most interesting to me. Specifically, the method replaces uninformative context samples with informative ones where "uninformative-ness" is measured by attention weights and "informative-ness" is measured by difficulty of classification.

**Weaknesses:**

* While the paper presents three innovations and claims that they are all tailored to multi-modal data, only the Bayesian aggregation is inherently related to multi-modal inputs. The dynamic context update and the RBF context mechanism do not utilize the multi-modal characteristic. This makes the contributions and claims less coherent.

* It seems the Bayesian aggregation mechanism is not ablated. It's not clear how much improvement it brings to the overall performance as it is the main mechanism responsible for tackling multi-modal inputs. Does the improved robustness come from the aggregation algorithm or the other two components?


**Questions:**

* Can the authors comment on the importance of the Bayesian aggregation component?
* It would be good to consolidate a single table (for one dataset) to directly show the effectiveness of each component in terms of accuracy, uncertainty, and robustness.

**Limitations:**

The work does not have an apparent negative societal impact.

---

> ### Author Rebuttal · Authors · 2023-08-09
>
> We thank the reviewer for the constructive comments. We would like to address the following statements.
>
> > “While the paper presents three innovations and claims that they are all tailored to multi-modal data, only the Bayesian aggregation is inherently related to multi-modal inputs. The dynamic context update and the RBF context mechanism do not utilize the multi-modal characteristic. This makes the contributions and claims less coherent.”
>
> Please note that our method is a joint neural process framework for uncertainty estimation of multimodal data. Although the framework consists of several components, all of them work coherently in an integrated way. Our proposal of dynamic context update and RBF enables neural processes to more accurately capture uncertainties, while the Bayesian aggregation makes neural processes able to deal with multimodal data. Our ablation study shows that the proposed framework is unable to work as effectively or efficiently as it does without any of the modules. From the perspective of multimodal uncertainty estimation via neural processes, we believe the innovations are coherent.
>
> > “It seems the Bayesian aggregation mechanism is not ablated. It's not clear how much improvement it brings to the overall performance as it is the main mechanism responsible for tackling multi-modal inputs. Does the improved robustness come from the aggregation algorithm or the other two components?” “Can the authors comment on the importance of the Bayesian aggregation component?”
>
> We thank the reviewer for the constructive suggestion. We conducted additional ablation studies on MBA with two baselines namely “Concat” and “Mean”. “Concat” bypasses MBA and directly provides $r^m_*$ of multiple modalities to the decoder (see Figure 1) by simple concatenation followed by passing to a MLP which lets $p(f(T^M_X)|C^M,T^M_X)$ in Equation (12) be parameterised by a decoder where $\\{C^M,T^M_X\\}=MLP(Concat(\\{r^m_*\\}^M\_{m=1}))$. $Concat(\\cdot)$ represents concatenating multiple vectors along their feature dimension. Similarly, “Mean” also bypasses MBA and simply averages the multiple modalities into single representation. Formally, $p(f(T^M_X)|C^M,T^M_X)$ parameterised by a decoder where $\\{C^M,T^M_X\\}=\\frac{1}{M} \\sum^M\_{m=1}r^m_*$. We conducted the same main experiments, and the results are shown below:
>
> Test accuracy ($\\uparrow$).
> \\begin{array}{ccccccc} \\hline {} & \\text{Handwritten} & \\text{CUB} & \\text{PIE} & \\text{Caltech101} & \\text{Scene15} & \\text{HMDB} \\\ \\hline \\text{Concat} & 99.35\\pm0.22 & 89.00\\pm1.24 & 89.71\\pm2.49 & 92.63\\pm0.18 & 77.18\\pm0.64 & 56.06\\pm2.13 \\\  \\text{Mean} & 99.45\\pm0.11 & 92.50\\pm2.43 & 90.88\\pm2.24 & 93.14\\pm0.25 & 77.60\\pm0.56 & 57.80\\pm1.97 \\\ \\text{MBA} &  \\mathbf{99.50\\pm0.00} & \\mathbf{93.50\\pm1.71} & \\mathbf{95.00\\pm0.62} & \\mathbf{93.46\\pm0.32} & \\mathbf{77.90\\pm0.71} & \\mathbf{71.97\\pm0.43} \\\ \\hline \\end{array}
>
> Test ECE ($\\downarrow$).
> \\begin{array}{ccccccc} \\hline {} & \\text{Handwritten} & \\text{CUB} & \\text{PIE} & \\text{Caltech101} & \\text{Scene15} & \\text{HMDB} \\\ \\hline \\text{Concat} & 0.007\\pm0.001 & 0.109±0.008 & 0.092±0.020 & 0.038±0.005 & 0.061±0.005 & 0.060±0.017 \\\ \\text{Mean} & 0.006±0.001 & 0.057±0.012 & 0.059±0.008 & 0.030±0.004 & 0.038±0.005 & 0.117±0.014 \\\ \\text{MBA} &  \\mathbf{0.005±0.001} & \\mathbf{0.049±0.008} & \\mathbf{0.040±0.005} & \\mathbf{0.017±0.003} & \\mathbf{0.038±0.009} & \\mathbf{0.028±0.006
> } \\\ \\hline \\end{array}
>
> Average test accuracy across 10 noise levels ($\\uparrow$).
> \\begin{array}{ccccccc} \\hline {} & \\text{Handwritten} & \\text{CUB} & \\text{PIE} & \\text{Caltech101} & \\text{Scene15} & \\text{HMDB} \\\ \\hline \\text{Concat} & 97.71±0.46 & 85.51±1.42 & 85.94±2.48 & 89.84±0.17 & 72.23±0.52 & 45.22±2.86 \\\ \\text{Mean} & 98.42±0.09 & 88.27±1.83 & 88.74±2.33 & 92.07±0.16 & 74.06±0.28 & 49.58±2.24 \\\ \\text{MBA} &  \\mathbf{98.58±0.10} & \\mathbf{88.96±1.98} & \\mathbf{93.80±0.49} & \\mathbf{92.83±0.18} & \\mathbf{74.14±0.35} & \\mathbf{64.11±0.15} \\\ \\hline \\end{array}
>
> Test accuracy ($\\uparrow$), ECE ($\\downarrow$), and OOD detection AUC ($\\uparrow$).
> \\begin{array}{ccccc} \\hline & \\text{Test accuracy} \\uparrow & \\text{ECE} \\downarrow & \\text{OOD AUC (SVHN)} \\uparrow & \\text{OOD AUC (CIFAR100)} \\uparrow \\\ \\hline \\text{Concat} & 74.24±0.27 & 0.125±0.005 & 0.781±0.016 & 0.728±0.004 \\\ \\text{Mean} & 74.72±0.24 & 0.109±0.003 & 0.803±0.007 & 0.742±0.003 \\\ \\text{MBA} & \\mathbf{74.92±0.07} & \\mathbf{0.011±0.001} & \\mathbf{0.872±0.002} & \\mathbf{0.786±0.005} \\\ \\hline\\end{array}
>
> In every case, MBA outperformed both baselines. While similar performance was observed for Handwritten, Scene15, and Caltech101, large differences were observed in CUB, PIE, and HMDB across different metrics. The test accuracy of CIFAR10 is almost consistent across all methods, but large gaps in ECE and OOD performance were observed. This highlights the importance of MBA, especially in robustness and calibration performance.

---

> > ### Comment · Reviewer_hoCV · 2023-08-18
> > **Response to Rebuttal**
> >
> > Thanks to the authors for adding the ablation study, this addressed my second question. But still, I think that the dynamic update and RBF context mechanism are not inherent progress towards multi-modality even though they do contribute to more accurate uncertainty estimation. Therefore, I keep my original score.

---

### Official Review · Reviewer_RNhe · 2023-07-02

**Soundness:** 3 good
**Presentation:** 2 fair
**Contribution:** 2 fair
**Rating:** 6
**Confidence:** 3

**Summary:**

This work proposes a confidence calibration algorithm for multimodal classification problems. The algorithm includes three key components: 1) dynamic context memory, 2) multimodal Bayesian aggregation, and 3) adaptive radial basis function. The algorithm is evaluated on multiple benchmarks using classification accuracy, calibration, and OOD performance. On multiple benchmarks, the proposed algorithm is able to obtain SOTA results.

**Strengths:**

1, The targeted problem and challenges are explicitly stated. For each challenge, a technically reasonable solution is proposed.
2, The proposed algorithm is able to obtain strong empirical results, obtaining SOTA results on multiple benchmarks and different metrics.
3, Comprehensive ablation studies have been conducted to demonstrate each components.
4, As illustrated in Figure 2, the proposed adaptive RBF algorithm outperformed other attention mechanisms.

**Weaknesses:**

1, Some notation definitions are confusing. Please refer to section "Questions" for details.
2, Because of the above notation section 3.2 is a bit hard to follow.
3, Ablation study is conducted for individual components. It would be interesting to design ablation studies and see how each component affects the end-to-end performance.
4, The component directly related to multimodality learning, i.e., multimodal Bayesian aggregation, is based on previously work [1]. It would be good if the authors can explain how the proposed MBA component is different from previous work, either in the revision or rebuttal.

[1] M. Volpp, F. Flürenbrock, L. Grossberger, C. Daniel, and G. Neumann. Bayesian context aggregation for neural processes. In International Conference on Learning Representations, 2021.

**Questions:**

1, **Line 71**, from my understanding, the context set C and target set T contains both training and test datasets. But $N_C +N_T=N_{\text{train}}$. Why is the summation of $N_C$ and $N_T$ only equal to the number of training points?

2, **Line 148 and 150**, what are $r^m$  and $s^m$? How are they connected/different from $r^m_{\*}$ and $s^m_{\*}$?

3, **Line 160**, how are encoders parameterized by $\theta$ and $\omega$ connected/different from those by $\phi$ and $\psi$?

**Limitations:**

The authors explicitly discussed the limitations.

---

> ### Author Rebuttal · Authors · 2023-08-09
>
> We thank the reviewer for the comments and questions. We would like to address the following questions.
>
> > "Line 71, from my understanding, the context set C and target set T contains both training and test datasets. But $N_C +N_T=N_{\\text{train}}$. Why is the summation of $N_C$ and $N_T$ only equal to the number of training points?"
>
> We would like to clarify that a context set and a target set are required for training and test datasets separately, This means that during training, the training dataset can be split to a context set and a target set (i.e., $N_C +N_T=N_{\\text{train}}$). Similarly, during testing, a context set and a target set are required. However, since we have no label information of the test dataset, the context set has to be given from somewhere else. A simple solution can be using the entire training set as the labelled context set, but it is highly inefficient especially for large datasets. Therefore, we have proposed dynamic context memory to store a smaller number of informative training samples during training that can be used in the inference stage.
>
> > “Line 148 and 150, what are $r^m$ and $s^m$? How are they connected/different from $r^m_*$ and $s^m_*$?”
>
> $r^m$ and $s^m$ are the encoded context representations which have no information about target samples. The attention mechanism measures the distance between the target samples and produces the weighted sum of $r^m$ and $s^m$ for specific target samples. These target-specific context representations are $r^m_*$ and $s^m_*$.
>
> > “Line 160, how are encoders parameterized by $\\theta$ and $\\omega$ connected/different from those by $\\phi$ and $\\psi$?”
>
> The encoders parameterised by $\\theta$ and $\\omega$ and those by $\\phi$ and $\\psi$ are used to approximate different distribution parameters. To be precise, $u^m$ encoded with $\\theta$ represents the mean of prior, $q^m$ encoded by $\\omega$ represents the diagonal covariance of prior, $r^m_*$ encoded by $\\phi$ represents the samples from $p({r^m_*}\_i \vert z_i)$, and  $s^m_*$ encoded by $\\psi$ represents the diagonal covariance of $p({r^m_*}\_i \vert z_i)$. Each of the encoders is implemented as an independent neural network.
>
> > “Ablation study is conducted for individual components. It would be interesting to design ablation studies and see how each component affects the end-to-end performance.”
>
> Due to the page limitation, the ablation studies were put in the appendix. We would like to highlight that we conducted all ablation studies which are outlined in Appendix C using the same experimental procedures including the datasets as our main experiments in order to show the effectiveness of each component.
>
> > “The component directly related to multimodality learning, i.e., multimodal Bayesian aggregation, is based on previously work [1]. It would be good if the authors can explain how the proposed MBA component is different from previous work, either in the revision or rebuttal.”
>
> Our proposed MBA has technical connections with the previous work [1]. Both methods are Bayesian approaches that aggregate multiple representations into a single one (latent variable $z$) by modelling the prior and likelihood of $z$ to obtain the posterior. However, there are significant differences between the previous work [1] and ours, regarding their intended applications and model specifications.
>
> The previous study [1] aimed to aggregate representations of different instances (or observations) into a single representation of latent variable $z$ for the context samples, using an uninformative prior on $z$. Since its primary purpose is to summarise context samples, it has no interactions with target samples.
>
> In contrast, our approach aggregates multiple modalities of the same instance (or an observation) into a single representation of latent variable $z$. We incorporate an informative prior $\\mathcal{N}(u^m,\\text{diag}(q^m)$ (Equation (6), L159, and Figure 1), derived from the context memory. Furthermore, by considering a target-specific representation ${r^m_*}\_i $ (the output of the proposed adaptive RBF attention) as a sample of $p({r^m_*}\_i \vert z_i)$, we model $p({r^m_*}\_i \vert z_i)$ as a distribution after observing a target sample, and $p(z_i)$ as a distribution prior to observing a target sample.

---

> > ### Comment · Reviewer_RNhe · 2023-08-19
> >
> > Thanks the authors for response. They have addressed all my questions and concerns. After considering the responses and the good empirical results, I decided to raise my score.

---

### Official Review · Reviewer_U6Gp · 2023-07-08

**Soundness:** 4 excellent
**Presentation:** 3 good
**Contribution:** 3 good
**Rating:** 7
**Confidence:** 4

**Summary:**

This paper proposes a multimodal neural processes (neural network generaliation of Gaussian processes) model.

The overall approach has several novel elements:
* A way to maintain a dynamic context set throughout training (e.g a support set for few-shot learning, these context sets are needed for neural processes)
* A Bayesian aggregation scheme for combining multiple modalities
* An adaptive RBF attention mechanism as an alternative to (vanilla) dot product attention which the authors argue is overconfident on OOD samples.

Experiments on a suite of datasets show that the proposed method is faster, more accurate and better calibrated, and better at detecting OOD samples than prior work.


**Strengths:**

This reads like a polished piece of work with clear writing, extensive ablations (in the supplementary) and novel technical contributions.  Experimental results seem fairly convincing (that the proposed method is indeed better than prior similar works).  My caveat is that my familiarity with this part of the literature on Neural Processes is passing at best.

**Weaknesses:**


I have no major complaints about this paper, but some things to point out are:
It is pretty unclear what are the actual multiple modalities in the datasets that are used for experiments (which makes these experiments not particularly compelling perhaps unless you are already familiar with this line of work).  I had to trace through a series of cited works to figure out that (likely) these different modalities are different features extracted e.g using different networks.
Section 3.2 could be written more clearly — I couldn’t figure out why there were multiple encoders until encountering Lemma 3.1 — it’d be much better to explain this up front.
A nit: it’s also not clear when reading section 3.1 how often the dynamic context memory is meant to be updated until I read the pseudocode much later (I’d recommend saying that it’s a per-minibatch update somewhere).


**Questions:**

See above.

**Limitations:**

Yes.

---

> ### Author Rebuttal · Authors · 2023-08-09
>
> We thank the reviewer for the constructive feedback. We would like to address the reviewer’s suggestions.
>
> > “It is pretty unclear what are the actual multiple modalities in the datasets that are used for experiments”
>
> We acknowledge that we have missed the details of the input modality in Appendix B.1. We will add the types of input modalities and the preprocessing steps for each dataset in our revised manuscript.
>
> > “Section 3.2 could be written more clearly — I couldn’t figure out why there were multiple encoders until encountering Lemma 3.1 — it’d be much better to explain this up front. A nit: it’s also not clear when reading section 3.1 how often the dynamic context memory is meant to be updated until I read the pseudocode much later (I’d recommend saying that it’s a per-minibatch update somewhere).”
>
> Thank you for the suggestion. We agree with the reviewer. We will elaborate why multiple encoders are necessary in the beginning of Section 3.2, and explain the per-minibatch update in the beginning of Section 3.1.

---

### Official Review · Reviewer_2g2g · 2023-07-17

**Soundness:** 3 good
**Presentation:** 2 fair
**Contribution:** 3 good
**Rating:** 5
**Confidence:** 3

**Summary:**

This paper proposes a new method for multimodal uncertainty estimation by extending neural processes. The authors summarize three challenges to do that and give solutions correspondingly. Experimental results show that the proposed method is more robust and outperforms existing baselines.

---

Thanks for the clarification! My main concerns are addressed.

**Strengths:**

The proposed method (MNPs) can achieve good performance empirically.

**Weaknesses:**

1. The motivation is not clear. Indeed, there is little discussion about neural processes with multimodal data, and extending unimodal neural processes to multimodal scenarios could be challenging. However, existing works discussed the multi-view data [16,21]. The reason why extending the neural processes for multimodal data (instead of extending/improving [16,21]) remains unclear.


2. Lack of understanding. I would like to suggest that the authors add some ablation studies. From current experiments, we can only see that proposed MNPs outperform baselines. However, why it can do that is not clear, e.g., why the unimodal method (DE) can achieve the best test accuracy.

**Questions:**

(Line 185): Maybe I missed something, but why using RBF can address the overconfident issue?

**Limitations:**

Yes

---

> ### Author Rebuttal · Authors · 2023-08-09
>
> Thank you for your time and comments. We would like to address the following points.
>
> > “However, existing works discussed the multi-view data [16,21]. The reason why extending the neural processes for multimodal data (instead of extending/improving [16,21]) remains unclear.”
>
> As we stated our motivation in L38-46, NPs bring the best of both worlds of GP, which has been shown to be well-calibrated and robust to domain-shift [21, 31, R1], and DNNs, which have representation power and efficiency. The current SOTA model [21] is a GP, which is robust but computationally expensive at the same time. We show in our experiments that we can achieve better or comparable performance with faster computational time (up to five folds). Also, in [21, R2] and our experiments, TMC [16] has been shown to have limited capability in calibration performance and OOD detection, which are essential uncertainty estimation downstream tasks [21, 31]. Therefore, existing methods are either less effective or less efficient, which motivate the development of our approach.
>
> > “I would like to suggest that the authors add some ablation studies. From current experiments, we can only see that proposed MNPs outperform baselines. However, why it can do that is not clear, e.g., why the unimodal method (DE) can achieve the best test accuracy.”
>
> Please refer to Appendix C for ablation studies comparing different context memory updating mechanisms, attention types, and adaptive learning of RBF attention. We would like to highlight that DE has achieved the best test accuracy for only one out of the seven datasets. In the other six datasets, our method outperformed DE. For uncertainty estimation, metrics like ECE, robustness to noise, and OOD detection performance are more important as they directly or indirectly quantify uncertainty estimation performance. For those metrics, our method outperformed DE in all seven datasets, which demonstrates the superior uncertainty estimation performance of our method.
>
> > “Why using RBF can address the overconfident issue?”
>
> We explained the reason in L186-191 of the paper. But please let us elaborate more here. The core component of RBF is the lengthscale controlling  the degree of smoothness in distance calculations. To illustrated, considering two points  $x$  and $x'$ where ${\\vert\\vert x-x' \\vert\\vert}^2=1$, the RBF value (indicative of closeness) can vary significantly based on the lengthscale: 0.98 if the lengthscale is 5 and 1.93e-22  if the lengthscale is 0.1. This underlines that an appropriately determined lengthscale can distinguish whether or not two samples come from different distributions. As highlighted in L203, this lengthscale parameter is often predetermined as a hyperparameter or an optimisable parameter that requires a complex initialisation. To overcome this limitation, we have proposed $\\mathcal{L}\_{RBF}$ to optimise the lengthscale without any complex initialisation. The effectiveness of this approach is demonstrated in Figure 2, Appendix C.2, and Appendix C.3.
>
> Additional references:
>
> [R1] S. G. Popescu, D. J. Sharp, J. H. Cole, K. Kamnitsas, and B. Glocker. Distributional gaussian process layers for outlier detection in image segmentation. In A. Feragen, S. Sommer, J. Schnabel, and M. Nielsen, editors, Information Processing in Medical Imaging, pages 415–427, Cham, 2021. Springer International Publishing. ISBN 978-3-030-78191-0.
>
> [R2] K. Zou, T. Lin, X. Yuan, H. Chen, X. Shen, M. Wang, and H. Fu. Reliable multimodality eye disease screening via mixture of student’s t distributions. arXiv preprint arXiv:2303.09790, 2023.
>
> [R3] Y. Ovadia, E. Fertig, J. Ren, Z. Nado, D. Sculley, S. Nowozin, J. Dillon, B. Lakshminarayanan, and J. Snoek. Can you trust your model's uncertainty? evaluating predictive uncertainty under dataset shift. In H. Wallach, H. Larochelle, A. Beygelzimer, F. d'Alché-Buc, E. Fox, and R. Garnett, editors, Advances in Neural Information Processing Systems, volume 32. Curran Associates, Inc., 2019.
>
> [R4] J. Mukhoti, A. Kirsch, J. van Amersfoort, P. H. Torr, and Y. Gal. Deterministic neural networks with appropriate inductive biases capture epistemic and aleatoric uncertainty. arXiv preprint arXiv:2102.11582, 2021.

---

### Official Review · Reviewer_fi6w · 2023-07-23

**Soundness:** 2 fair
**Presentation:** 2 fair
**Contribution:** 3 good
**Rating:** 6
**Confidence:** 3

**Summary:**

This paper extends one of the promising uncertainty estimation method: Neural Process from unimodal to multimodal. This is motivated by the fact that: current techniques are predominantly designed for unimodal data, and directly applying them to multimodal information is ineffective. However, this extension poses several challenges: 1) how fuse information from different modalities effectively and efficiently; 2) the context memory of the original design grows in proportion to M - the number of modality, which is memory-consuming, and sets the question of how to maintain it small yet informative; 3) how to find an appropriate length-scale of the Radial Basis Function (RBF) that ensures a tight boundary between in-distribution and out-of-distribution data.
To tackle these challenges, the paper introduces three solutions: 1) Dynamic Context Memory to choose the most informative samples in the context memory, 2) Multimodal Bayesian Aggregation, and 3) Adaptive RBF Attention. Empirical experiments show that the method can outperform other unimodal and multimodal baselines in terms of maintaining good accuracy and calibration at the presence of noise and in terms of out-of-distribution detection.

**Strengths:**

1. originality: the first to extend Neural Process from unimodal to multimodal and effectively combine existing techniques to solve some of the challenges incurred.
2. clarity: in general, the paper articulates its ideas clearly, even though some parts remain difficult to understand or unclear in terms of details (see the weakness part)
3. significance: the proposed method shows promising results in terms of outperform most unimodal and multimodal baselines.

**Weaknesses:**

**Clarity**: the clarity of the technical introduction (Sec. 3) and the experiment (Sec. 4) parts can be largely improved.
1. The method introduction part could be **challenging to comprehend for readers unfamiliar with the Neural Process and its technical details**. It lacks adequate contextual knowledge, such as an explanation for why the context memory is required, why it is used to store training samples rather than using a context feature. I think the Sec. 2 should be improved to be more comprehensive and general in introducing the Neural Process, to avoid the readers to revisit several referenced papers repeatedly.
1. The **math details can be overwhelming and at times confusin**g. For instance, when reviewing Sec 3.3, it necessitates frequent referral back to Section 3.2. It would be helpful if the semantic meaning of symbols is reintroduced when they appear, preventing the need for readers to revisit previous sections for notation understanding.
	1. **Derivation details**: The logic in some derivations is not easy to understand, e.g. Line 221 introduces the function $f(T_X^M)$ without any prior explanation. Also, there is no explanation on how equation (12) is derived, and why softmax is applied on a density function. These issues impede reading and understanding. The formula (4) and formula (6) are not consistent as their parameter notation is not the same on $\omega$ and $\psi$.
	2. The MNPs **pseudocode** lacks a high-level introduction and has heavy reference links, failing to provide clarifying information.
	3. **Captions**: Some table captions lack sufficient information. For instance, Table 4/5 does not clearly describe the dataset and settings used.
	4. **Experimental Setup**: The presentation of the experimental settings, such as how datasets e.g. Caltech101 were modified into multi-view datasets, is not clear.

**Soundness**: Despite the multimodal setting, the experiments use only multi-view image data. The lack of experiments with multimodal combinations such as image + text impedes the proof of the model's effectiveness in various settings.


**Questions:**

see questions listed on the weakness part

**Limitations:**

The author have listed one limitation that the updating mechanism is not theoretically guaranteed to obtain the optimal context memory.

---

> ### Author Rebuttal · Authors · 2023-08-09
>
> We thank the reviewer for the constructive feedback and comments. We would like to address the following statements.
>
> > "It lacks adequate contextual knowledge, such as an explanation for why the context memory is required, why it is used to store training samples rather than using a context feature."
>
> We will provide more explanations in the revision as suggested. Here we would like to clarify why context memory is required. NPs require a labelled context set during both training and inference in order to make predictions for a target set. During training, training samples, which are labelled, can be used as the context set. However, during the inference stage, the labelled context set is not given for our task. Thus, a context memory which stores labelled training samples is crucial to enable an efficient and effective inference.
>
> The context memory $\\{ C_X,C_Y \\}$ stores training samples in input space so that when a target sample $T_X$ is given, the distance between $C_X$ and $T_X$ is measured to weight the context elements for generating target-specific context representations in a non-parametric way (i.e., weighted sum of encoded context samples for a specific target sample). It is worth noting that, as we stated in L65, we consider the input space to be a feature space.
>
> > “I think the Sec. 2 should be improved to be more comprehensive and general in introducing the Neural Process, to avoid the readers to revisit several referenced papers repeatedly…It would be helpful if the semantic meaning of symbols is reintroduced when they appear, preventing the need for readers to revisit previous sections for notation understanding.”
>
> We appreciate your suggestion. We will include a general introduction of NPs at Section 2 in the revised manuscript.
>
> > “The logic in some derivations is not easy to understand, e.g. Line 221 introduces the function $f(T^M_X)$ without any prior explanation. Also, there is no explanation on how equation (12) is derived, and why softmax is applied on a density function.”
>
> Thank you for pointing this out. Please let us provide a detailed explanation here. In Gaussian process classification such as [60, 21, 38, 18] where the likelihood  is categorical, class probability is obtained in two stages. First, the predictive distribution is acquired in the form of a Gaussian distribution, which is then sampled to squash through the softmax function. Similarly in NPs, the predictive distribution is acquired as a Gaussian distribution $p(T_Y\\vert T_X,C_X,C_Y)$ which is suitable for regression but not for classification. Thus, we introduced a latent function $f(\\cdot)$ to estimate the latent distribution $p(f(T^M_X)|C^M,T^M_X)$ which is then sampled to squash through the softmax function to estimate the class probability.
>
> > “The formula (4) and formula (6) are not consistent as their parameter notation is not the same on $\\omega$ and $\\psi$.”
>
> We would like to clarify that the parameters $\\phi$, $\\psi$, $\\theta$, and $\\omega$ are meant to be different, indicating four distinct encoders. The reasons for using different encoders are that $ r_m^* $ for $\\phi$, $s_m^*$ for $\\psi$, $u^m$ for $\\theta$, and $q^m$ for $\\omega$ are used to approximate different distribution parameters: $ r_m^* $  as samples from $p({r^m_*}\_i \vert z_i)$, $s_m^*$ as diagonal covariance of $p({r^m_*}\_i \vert z_i)$, $u^m$ as mean of prior, and $q^m$ as diagonal covariance of prior.
>
> > “The MNPs pseudocode lacks a high-level introduction and has heavy reference links, failing to provide clarifying information.”
>
> We agree with the reviewer. We will add short comments for each line to make it more interpretable.
>
> > “Captions: Some table captions lack sufficient information. For instance, Table 4/5 does not clearly describe the dataset and settings used.”
>
> We kept captions as concise as possible due to the space constraints. Please refer to Section 5.1, 5.2, and Appendix B for experimental details.
>
> > “Experimental Setup: The presentation of the experimental settings, such as how datasets e.g. Caltech101 were modified into multi-view datasets, is not clear.” “Despite the multimodal setting, the experiments use only multi-view image data. The lack of experiments with multimodal combinations such as image + text impedes the proof of the model's effectiveness in various settings.”
>
> Thank you for bringing this to our attention. In Appendix B.1, we provided an outline of the dataset details but missed information about the types of input modalities and the preprocessing. We will add these to our revised manuscript.
>
> In summary, the datasets consist of a range of input modalities. Handwritten, PIE, HMDB, Scene15, Caltech101, CIFAR10, SVHN, and CIFAR100 are image datasets with diverse features associated with each image. This diversity comes from either different feature extraction methods or augmentation to generate multiple modalities from a single image. Additionally, CUB represents an image+text dataset.
>
> It's important to clarify that we consider multimodality to involve multiple inputs regardless of whether these inputs share the same type (e.g., image+image) or differ in their types (e.g., image+text).

---

> > ### Comment · Reviewer_fi6w · 2023-08-15
> >
> > Thanks for the responses to my and other reviewers' questions.
> >
> > The major concern I have now is about the experiment part. In my perspective, the most important application of multimodal uncertainty estimation lies in **integrating information from different sources (e.g. texts+images, images from different cameras)** to gauge uncertainty. For instance, in the scenario where image data might be highly blurry and the prediction based on the image can be uncertain, but the text data has predictive information, by leveraging both, one can achieve a more accurate measure of uncertainty.
> >
> > Using feature extraction methods and augmentations to procure multiple inputs fails to capture the real world cases, given they are produced from the same source of information, especially if this feature extraction methods are implemented using different random seeds. If there are many redundant information in every modality, it's challenging to assert that improvements in the uncertainty estimator would generalize effectively across diverse data sources.
> >
> > Based on this concern, I decide to remain my score.

---

> > > ### Author Response · Authors · 2023-08-16
> > >
> > > Dear reviewer,
> > >
> > > We appreciate your feedback and agree on your comment "For instance, in the scenario where image data might be highly blurry and the prediction based on the image can be uncertain, but the text data has predictive information, by leveraging both, one can achieve a more accurate measure of uncertainty." This is exactly what our method aims to accomplish. We also agree with you on the redundant information of multiple modalities due to the same source of input.
> > >
> > > Please note that one of our datasets, CUB, contains both textual descriptions and images. Its experimental results on robustness to noisy samples in section 5.1 demonstrate the desirable property which was mentioned above.
> > >
> > > Nevertheless, to make our evaluation even more comprehensive, we are currently conducting additional experiments with a dataset consisting of multiple modalities from different input sources. We will update our response as soon as the experiments are finished.
> > >
> > > Kind regards,
> > >
> > > The authors

---

> > > > ### Author Response · Authors · 2023-08-17
> > > >
> > > > Dear Reviewer,
> > > >
> > > > We have conducted additional experiments on the Oxford-102 Flower dataset [R1] which consists of images and captions. As the images and the captions are formulated from different input sources, it reflects a real-world scenario where input modalities are less redundant. We leveraged Inception v3 [50] and doc2vec [R2] to extract visual and textual features and used samples belonging to the first ten classes (3,485 samples in total) for this experiment. We split the dataset into a training set and a test set with 0.7:0.3 ratio and set the number of context samples/inducing points for ETP, MGP, and MNP to 200. We will include more detailed experimental settings in the revised manuscript.
> > > >
> > > > The same experimental procedure of the main experiment in section 5.1 was conducted, and the results are as follows:
> > > >
> > > > \\begin{array}{cccc} \\hline {} & \\text{Test accuracy} \\uparrow & \\text{Test ECE} \\downarrow & \\text{Average test accuracy across 10 noise levels} \\uparrow \\\ \\hline \\text{MCD} & 94.86±1.44 & 0.195±0.012 & 79.65±0.33 \\\  \\text{DE(EF)} & 98.29±0.80 & 0.087±0.007 & \\underline{88.65±0.3} \\\ \text{SNGP} & 93.14±1.04 & 0.454±0.022 & 69.00±1.17 \\\ \\text{ETP} & 98.10±0.95 & 0.068±0.014 & 82.09±0.85 \\\ \\text{DE(LF)} & 96.76±1.09 & 0.394±0.018 & 83.40±1.05 \\\ \\text{TMC} & 94.67±1.09 & 0.123±0.008 & 81.40±0.84 \\\ \\text{MGP} & \\mathbf{98.67±0.52} & \\underline{0.037±0.008} & 76.55±0.55 \\\ \\text{MNP (Ours)} & \\underline{98.48±1.09} & \\mathbf{0.017±0.005} & \\mathbf{94.19±0.42} \\\ \\hline \\end{array}
> > > >
> > > > While there is a marginal difference in test accuracy of DE(EF), ETP, MGP, and MNP, a large gap in test ECE and average test accuracy with noisy samples was observed. This illustrates MNP’s robustness and calibration performance that outperform other baselines. More importantly, MNP is able to maintain test accuracy under noisy conditions with a slight decrease in accuracy (98.48->94.19). Other baselines have shown significant decrease in accuracy (e.g., MGP: 98.67->76.55 or ETP: 98.10->82.09). This experiment, along with CUB, shows the effectiveness of MNP's uncertainty estimation performance across diverse input modalities.
> > > >
> > > > Additional references:
> > > >
> > > > [R1] S. Reed, Z. Akata, X. Yan, L. Logeswaran, B. Schiele, and H. Lee. Generative adversarial text to image synthesis. In M. F. Balcan and K. Q. Weinberger, editors, Proceedings of The 33rd International Conference on Machine Learning, volume 48 of Proceedings of Machine Learning Research, pages 1060–1069, New York, New York, USA, 20–22 Jun 2016. PMLR.
> > > >
> > > > [R2] Q. Le and T. Mikolov. Distributed representations of sentences and documents. In E. P. Xing and T. Jebara, editors, Proceedings of the 31st International Conference on Machine Learning, volume 32 of Proceedings of Machine Learning Research, pages 1188–1196, Bejing, China, 22–24 Jun 2014. PMLR.
> > > >
> > > > Kind regards,
> > > >
> > > > The authors

---

### Author Rebuttal · Authors · 2023-08-09

**Continued Rebuttal for Reviewer hoCV**

> “It would be good to consolidate a single table (for one dataset) to directly show the effectiveness of each component in terms of accuracy, uncertainty, and robustness.”

Thank you for the suggestion. We would like to provide the first and the last datasets which are Handwritten and CIFAR10 for the comparison.

Handwritten.
\\begin{array}{c|ccc|ccc|ccc} \\hline & {} & \\text{Context memory} & {} & {} & \\text{Attention} &  {} & {} & \\text{Multimodal aggregation}  & {} \\\ {} & \\text{Random} & \\text{FIFO} & \\text{MSE (Ours)} & \\text{Dot Softmax} & \\text{RBF Softmax} & \\text{RBF Sparsemax (Ours)} & \\text{Concat} & \\text{Mean} & \\text{MBA (Ours)} \\\ \\hline \text{Test accuracy} \\uparrow & 99.40±0.14 & 99.30±0.11 & \\mathbf{99.50±0.00} & 99.00±0.18 & 98.80±0.45 & \\mathbf{99.50±0.00} & 99.35±0.22 & 99.45±0.11 & \\mathbf{99.50±0.00} \\\ \text{ECE} \\downarrow& 0.007±0.001 & 0.007±0.001 & \\textbf{0.005±0.001} & 0.008±0.003 & 0.019±0.005 & \\textbf{0.005±0.001} & 0.007±0.001 & 0.006±0.001 & \\textbf{0.005±0.001} \\\ \\text{Accuracy with noisy inputs} \\uparrow & 98.39±0.21 & 98.51±0.11 & \\mathbf{98.58±0.10} & 77.99±0.32 & 94.56±0.66 & \\mathbf{98.58±0.10} & 97.71±0.46 & 98.42±0.09 & \\mathbf{98.58±0.10} \\\ \\hline\\end{array}

CIFAR10.
\\begin{array}{c|ccc|ccc|ccc} \\hline & {} & \\text{Context memory} & {} & {} & \\text{Attention} &  {} & {} & \\text{Multimodal aggregation}  & {} \\\ {} & \\text{Random} & \\text{FIFO} & \\text{MSE (Ours)} & \\text{Dot Softmax} & \\text{RBF Softmax} & \\text{RBF Sparsemax (Ours)} & \\text{Concat} & \\text{Mean} & \\text{MBA (Ours)} \\\ \\hline \text{Test accuracy} \\uparrow & 74.61±0.22 & 74.82±0.11 & \\mathbf{74.92±0.07} & 68.81±0.62 & 67.65±0.16 & \\mathbf{74.92±0.07} & 74.24±0.27 & 74.72±0.24 & \\mathbf{74.92±0.07} \\\ \text{ECE} \\downarrow& 0.073±0.005 & 0.073±0.006 & \\mathbf{0.011±0.001} & 0.130±0.019 & 0.080±0.001 & \\mathbf{0.011±0.001} & 0.125±0.005 & 0.109±0.003 & \\mathbf{0.011±0.001} \\\ \\text{OOD SVHN} \\uparrow & 0.860±0.003 & 0.862±0.007 & \\mathbf{0.872±0.002} & 0.849±0.009 & 0.864±0.006 & \\mathbf{0.872±0.002} & 0.781±0.016 & 0.803±0.007 & \\mathbf{0.872±0.002} \\\ \\text{OOD CIFAR100} \\uparrow & 0.777±0.002 & 0.778±0.005 & \\mathbf{0.786±0.005} & 0.775±0.005 & 0.771±0.006 & \\mathbf{0.786±0.005} & 0.728±0.004 & 0.742±0.003 & \\mathbf{0.786±0.005} \\\ \\hline\\end{array}

By examining the two tables, it becomes clear that every component has an impact on all metrics. Nevertheless, noticeable differences, particularly in test accuracy, ECE, and robustness to noise, can be observed across different attention types. Notably, when it comes to OOD performance, the multimodal aggregation methods show more substantial differences.

---

### Decision · Program_Chairs · 2023-09-21

**Decision:**

Accept (poster)

**Comment:**

This paper looks at uncertainty estimation and out-of-distribution detection (OOD) in the context of multi-modal models. Specifically, the works looks at it from the perspective of neural processes (NPs) that maintain a dynamic context memory, and proposes a multimodal Bayesian aggregation method as well as adaptive radial basis functions. Results are demonstrated with respect to calibration performance, robustness under shift, and OOD detection for a set of multi-modal datasets including those with heterogeneous features or images+text as different modalities.

  The reviewers appreciated the novelty and significance of the proposed method, and mentioned a number of weaknesses including: 1) Lack of motivation, context, and detail for those not familiar with NPs (fi6w, 2g2g), 2) Overall lack of details, precision, and consistent notation for the derivation portions (fi6w, RNhe), 3) Description of what modalities are covered by the datasets (fi6w, U6Gp) and motivation of the method w.r.t to setting (hoCV), and 4) Lack of understanding and more thorough analysis about why the method performs well (2g2g, hoCV).

  In responses, the authors provided a rebuttal, including new experiments (Oxford-102 with images and captions, as well as new ablations) or pointers to those in the appendix. In the end, several reviewers mentioned that they were satisfied with the response, at least partially, and kept their overall positive score or increased it. After considering all of the materials (paper, reviews, and rebuttal) I believe the paper does make a contribution to an interesting underexplored area of uncertainty estimation for multi-modal datasets. The method is well-motivated and the results improve upon the state of the art. As a result, I recommend acceptance.